# Cytochalasin B-Induced Membrane Vesicles from Human Mesenchymal Stem Cells Overexpressing IL2 Are Able to Stimulate CD8^+^ T-Killers to Kill Human Triple Negative Breast Cancer Cells

**DOI:** 10.3390/biology10020141

**Published:** 2021-02-10

**Authors:** Daria S. Chulpanova, Zarema E. Gilazieva, Sevindzh K. Kletukhina, Aleksandr M. Aimaletdinov, Ekaterina E. Garanina, Victoria James, Albert A. Rizvanov, Valeriya V. Solovyeva

**Affiliations:** 1Institute of Fundamental Medicine and Biology, Kazan Federal University, 420008 Kazan, Russia; DaSChulpanova@kpfu.ru (D.S.C.); ZarEGilazieva@kpfu.ru (Z.E.G.); SKKletuhina@kpfu.ru (S.K.K.); AMAjmaletdinov@kpfu.ru (A.M.A.); EEGaranina@kpfu.ru (E.E.G.); Albert.Rizvanov@kpfu.ru (A.A.R.); 2Biodiscovery Institute, School of Veterinary Medicine and Science, University of Nottingham, Nottingham LE12 5RD, UK; Victoria.James@nottingham.ac.uk

**Keywords:** immunotherapy, mesenchymal stem cells, extracellular vesicles, cytochalasin B, interleukin 2, immune cell activation, cancer therapy, human triple negative breast cancer

## Abstract

**Simple Summary:**

Almost all human cells release extracellular vesicles participating in intercellular communication. Extracellular vesicles are rounded structures surrounded by the cytoplasmic membrane, which embody cytoplasmic contents of the parental cells, which makes extracellular vesicles a promising therapeutic tool for cell-free cancer therapy. In this study, human mesenchymal stem cells were genetically modified to overexpress human interleukin-2 (IL2), a cytokine which regulates the proliferation and activation of immune cells. Membrane vesicle release from native and genetically modified stem cells was induced by cytochalasin B treatment to increase the yield of membrane vesicles. To evaluate the immunomodulating properties of isolated membrane vesicles, immune cells were isolated from human peripheral blood and co-cultured with membrane vesicles from native or IL2 overexpressing stem cells. To analyze the anti-tumor activity of immune cells after interaction with IL2-enriched membrane vesicles, immune cells were co-cultured with triple negative breast cancer cells. As a result, IL2-enriched membrane vesicles were able to activate and stimulate the proliferation of immune cells, which in turn were able to induce apoptosis in breast cancer cells. Therefore, the production of IL2-enriched membrane vesicles represents a unique opportunity to meet the potential of extracellular vesicles to be used in clinical applications for cancer therapy.

**Abstract:**

Interleukin 2 (IL2) was one of the first cytokines used for cancer treatment due to its ability to stimulate anti-cancer immunity. However, recombinant IL2-based therapy is associated with high systemic toxicity and activation of regulatory T-cells, which are associated with the pro-tumor immune response. One of the current trends for the delivery of anticancer agents is the use of extracellular vesicles (EVs), which can carry and transfer biologically active cargos into cells. The use of EVs can increase the efficacy of IL2-based anti-tumor therapy whilst reducing systemic toxicity. In this study, human adipose tissue-derived mesenchymal stem cells (hADSCs) were transduced with lentivirus encoding IL2 (hADSCs-IL2). Membrane vesicles were isolated from hADSCs-IL2 using cytochalasin B (CIMVs-IL2). The effect of hADSCs-IL2 and CIMVs-IL2 on the activation and proliferation of human peripheral blood mononuclear cells (PBMCs) as well as the cytotoxicity of activated PBMCs against human triple negative cancer MDA-MB-231 and MDA-MB-436 cells were evaluated. The effect of CIMVs-IL2 on murine PBMCs was also evaluated in vivo. CIMVs-IL2 failed to suppress the proliferation of human PBMCs as opposed to hADSCs-IL2. However, CIMVs-IL2 were able to activate human CD8^+^ T-killers, which in turn, killed MDA-MB-231 cells more effectively than hADSCs-IL2-activated CD8^+^ T-killers. This immunomodulating effect of CIMVs-IL2 appears specific to human CD8^+^ T-killer cells, as the same effect was not observed on murine CD8^+^ T-cells. In conclusion, the use of CIMVs-IL2 has the potential to provide a more effective anti-cancer therapy. This compelling evidence supports further studies to evaluate CIMVs-IL2 effectiveness, using cancer mouse models with a reconstituted human immune system.

## 1. Introduction

Cytokine-based immunotherapy is frequently used in cancer treatment. Interleukin 2 (IL2) and interferons (IFNs) were the first cytokines used for cancer treatment due to their anti-tumor effects, including inhibition of tumor cell growth and differentiation [1]. The anti-cancer effect of high-dose (HD) IL2 therapy was demonstrated by West et al., where two complete and five partial responses were reported in 23 patients with metastatic renal cell carcinoma [2]. The therapeutic efficacy of low-dose IL2 treatment was demonstrated by Storter et al., where the objective response was reported in 17% of patients with metastatic renal cell carcinoma [3]. In more recent clinical trials, the efficacy of IL2 therapy has been shown in renal cancer patients with “good” and “poor” prognoses [4,5]. Based on current data, the United States Food and Drug Administration (FDA) approval has been granted for the use of HD IL2 therapy for the treatment of renal cell carcinoma and metastatic melanoma [6].

IL2 is predominantly produced by antigen-stimulated CD4^+^ T cells [7], as well as natural killer (NK) cells, NKT-cells, CD8^+^ T-cells, mast cells and dendritic cells (DCs) [8,9,10,11]. IL2 can stimulate the cytotoxic activity of CD8^+^ T-cells and NK cells [12] by binding to the IL2 receptor (IL2R) and activating several intracellular signalling pathways, including the Janus kinase (JAK) signal transducer, activator of the transcription (STAT) pathway, the phosphoinositide 3-kinase (PI3K) AKT pathway and the mitogen-activated protein kinase (MAPK) pathway [7].

The therapeutic potential of combining IL2 with other immune modulating treatments to treat different malignancies is currently under investigation. For example, human colon cancer cells expressing IL2 and IFN-β failed to establish tumors in animal models [13]. Furthermore, the combination of HD IL2 and CAR-T-cell therapy has demonstrated an objective response in six out of eight patients with advanced B-cell malignancies [14]. However, despite this strong therapeutic potential, recombinant IL2 immunotherapy faces several problems, including a short half-life and systemic side effects. The half-life of IL2 is a few minutes, with the compound being immediately eliminated from the circulation by the kidneys [15]. Whilst HD IL2 could extend the length of time the cytokine persists within the circulation, the higher dose may also lead to increased vascular permeability, hypotension, pulmonary edema, hepatocyte cell death and kidney failure [16]. In addition, prolonged IL2 exposure can stimulate regulatory T-cells (Tregs), inhibiting the tumor-targeting immune response [17]. For these reasons, the therapeutic use of IL2 is currently limited to patients with advanced renal cell carcinoma and melanoma. However, recent advances in our understanding of the mechanisms of cytokine anti-tumor effects has potentially opened new avenues for the clinical use of IL2. These include combining IL2 and Treg inhibitors, such as anti-cytotoxic T-lymphocyte-associated protein 4 (CTLA-4) and anti-programmed cell death protein 1 (PD-1) [18]. The combination of IL12 [19] and IL21 [20] has also been shown to block IL2-induced Treg activation, whilst changing the administration of IL2 from a single HD to a subcutaneous low-dose has been shown to prevent vascular permeability and hypotension [21].

Additional strategies to reduce the systemic toxicity of therapeutic agents include the use of mesenchymal stem cells (MSCs) and MSC-derived extracellular vesicles (EVs) as vehicles for targeted delivery [22,23]. For example, injection of exosomes derived from miR-146-expressing MSCs into primary brain tumor rat models caused a significant decrease in tumor growth [24]. EVs isolated from MSCs overexpressing tumor necrosis factor ligand superfamily member 10 (TRAIL) induced apoptosis in 11 cancer cell lines in a dose-dependent manner, but showed no cytotoxicity in human bronchial epithelial cells in vitro [25]. Therefore, the use of MSC-derived EVs as non-cell delivery systems, in place of MSCs themselves, not only promotes an effective response, but also avoids the risk of unlimited cell growth, undesirable transformation, and potential tumor formation [23]. Large-scale production of EVs for clinical use is not without its challenges, the most prominent of which is achieving a sufficient yield of EVs when endogenously produced [26]. Alternative approaches are being applied to increase EV yields, one of which is the use of cytochalasin B. Cytochalasin B causes actin filament dissociation and the shaking of such cytochalasin B-treated cells causes cell disintegration and the formation of multiple vesicles that are built from cell plasma membrane [27]. Use of cytochalasin B-induced membrane vesicle (CIMV) production offers a new approach to simplify the isolation of vesicles on a large scale to enable their use in clinical practice. To reflect the potential clinical use of IL2-containing EVs as a cancer therapeutic, we have used cytochalasin B to produce IL2 containing EVs from human adipose tissue-derived mesenchymal stem cells (hADSCs) in this study. The therapeutic potential of CIMVs carrying IL2 (CIMVs-IL2) was compared to hADSCs overexpressing IL2, to evaluate their effects on the proliferation and activation of immune cells both in vitro and in vivo.

## 2. Materials and Methods

### 2.1. Cells and Culture Conditions

Adipose tissue and blood samples were collected from two donors at the Republican Clinical Hospital in accordance with approved ethical standards and current legislation (the protocol was approved by the Committee on Biomedical Ethics of Kazan Federal University (No. 3, 03/23/2017)). Informed consent was obtained from each donor.

hADSCs were isolated from two different donors according to the previously developed protocol [28]. Briefly, subcutaneous adipose tissue samples were obtained during liposuction procedures. Lipoaspirate was washed three times with phosphate buffered saline (PBS) and treated with 0.2% hepatopancreas collagenase solution (Biolot, St. Petersburg, Russia) at 37 °C for 1 h. Red blood cells were lysed using a RBC Lysis Buffer (BioLegend, San Diego, CA, USA). The isolated cells were stained with Trypan Blue Solution (0.4%, Gibco, Grand Island, NY, USA) to determine the viability. Cells were cultured in a DMEM/F12 medium (PanEco, Moscow, Russia) supplemented with 10% fetal bovine serum (FBS, Invitrogen, Waltham, MA, USA), 2 mM of L-glutamine and antibiotics (100 U/mL of penicillin, 100 μg/mL of streptomycin, Biolot, St. Petersburg, Russia) and incubated at 37 °C in a 5% CO_2_/95% humidified air incubator. hADSCs were used up to passages 5–7.

Peripheral blood mononuclear cells (PBMCs) from four different donors were isolated using the Ficoll–Paque density gradient centrifugation as previously described [29]. Isolated PBMCs were cultured in a RPMI-1640 medium (PanEco, Moscow, Russia) with 10% FBS (HyClone, Logan, UT, USA), 2 mM of L-glutamine and antibiotics (PanEco, Moscow, Russia) and were incubated at 37 °C, 5% CO_2_/95% humidified air.

Human triple negative MDA-MB-231 breast cancer cells (#HTB-26), human triple negative MDA-MB-436 breast cancer cells (#HTB-130) and 293T embryonic kidney cells (#CRL-3216) were obtained from the American Type Culture Collection (ATCC, Manassas, VA, USA). MDA-MB-231 and 293T cells were cultured in a DMEM medium (PanEco, Moscow, Russia) with 4500 mg/L of glucose, 10% FBS (HyClone, Logan, UT, USA), 4 mM of L-glutamine and antibiotics (PanEco, Moscow, Russia) and were incubated at 37 °C, 5% CO_2_/95% humidified air. MDA-MB-436 cells were cultured in a RPMI-1640 medium (10% FBS, 2 mM of L-glutamine and antibiotics) (PanEco, Moscow, Russia). Cells were maintained according to standard protocols. Cell morphology was examined by an Axio Observer.Z1 (CarlZeiss, Jena, Germany) microscope and Axio Vision Rel. 4.8 software.

### 2.2. Differentiation of hADSCs into Adipocytes, Chondrocytes and Osteoblasts

To investigate the differentiation ability of native hADSCs, hADSCs-BFP and hADSCs-IL2 g, a StemPro^®^ Adipogenesis Differentiation Kit (#A10070-01), StemPro^®^ Chondrogenesis Differentiation Kit (#A10071-01) and StemPro^®^ Osteogenesis Differentiation Kit (#A10072-01, all Gibco, Grand Island, NY, USA) were used according to the manufacturer’s instructions. To induce adipogenic differentiation, hADSCs (1 × 10^4^ cells) were seeded in a 24-well plate. hADSCs were allowed to adhere, and then the culture medium was replaced with the differentiation medium. During differentiation, the medium was replaced every 3–4 days. After 14 days, to demonstrate adipogenic differentiation, cells were fixed (10% paraformaldehyde, 10 min, RT) and stained with 0.3% oil red O (#O0625, Sigma–Aldrich, St. Louis, MO, USA). To induce chondrogenic differentiation, hADSCs (5 × 10^4^ cells in 5 μL) were seeded in the center of the well in 24-well plates and incubated for 3 h. After that, chondrogenesis media was added. After 21 days, the presence of chondrogenic proteins was examined by staining fixed cells (10% paraformaldehyde, 10 min, RT) with Alcian blue (1% in 0.1N HCl; #A5268, Sigma–Aldrich, St. Louis, MO, USA). To induce osteogenic differentiation, hADSCs (1 × 10^4^ cells) were seeded in 24-well plates. hADSCs were allowed to adhere, and then the culture medium was replaced with the differentiation medium. During differentiation, the medium was replaced every 3–4 days. After 28 days, cells were fixed with 10% paraformaldehyde for 10 min at RT and stained with 1% aqueous AgNO_3_ for 1 h (in darkness) to determine the formation of calcified extracellular matrix.

### 2.3. Lentivirus Production

Lentiviral vector plasmid-encoding human IL2 gene (pLX304-IL2) was obtained from the Harvard Plasmid Database (#HsCD00421565-4). Lentiviral plasmid pLX303-BFP-encoding blue fluorescent protein (BFP) gene was produced by sub-cloning of the BFP gene, which was sub-cloned from the donor vector (pDONR221) into the lentiviral plasmid vector pLX303 (#25897, Addgene, Watertown, MA, USA) by LR recombination using the Gateway™ LR Clonase™ II Enzyme mix (#11791020, Invitrogen, Waltham, MA, USA) according to the manufacturer’s recommendations.

Production of lentiviruses, the second-generation replication-incompetent lentiviruses (LVs), encoding IL2 or BFP genes, was performed by calcium phosphate transfection co-transfection of 293T cells with three plasmids encoding: expression plasmid (pLX304-IL2 or pLX303-BFP); pCMV-dR8.2 dvpr packaging plasmid (#8455, Addgene, Watertown, MA, USA); and pCMV-VSV-G enveloping plasmid (#8454, Addgene, Watertown, MA, USA) [30]. The concentration of resulting lentiviruses LV-IL2 and LV-BFP using ultracentrifugation was performed for 2 h at 26,000 rpm at 4 °C. The viral titer was determined by infecting cells at various dilutions of the viral stock and determining the percentage of transduced cells by flow cytometry.

### 2.4. Genetic Modification and Selection

Native hADSCs were plated at a concentration of 5 × 10^4^ cells in 6-well culture plates and incubated for 24 h. After that, LV-IL2 or LV-BFP in serum-free DMEM/F12 were added to the cells at a multiplicity of infection (MOI) of 10. The cells were incubated with viruses for 6 h and then the medium was aspirated and completely replaced with a fresh one. To select transduced cells, the cells were cultured with blasticidin S (5 μg/mL, Invitrogen, Waltham, MA, USA) for 10 days.

### 2.5. Isolation of Cytochalasin B-Induced Membrane Vesicles

Artificial membrane vesicles were isolated from native and genetically modified hADSCs using cytochalasin B. For this, hADSCs were trypsinized, washed with PBS, and resuspended in a serum-free DMEM/F12 medium at a concentration of 2 × 10^5^ cells/mL. After that, 1.5 μL of cytochalasin B stock solution (10 mg/mL) (cytochalasin B from *Drechslera dematioidea*, #C6762-5MG, Sigma-Aldrich, St. Louis, MO, USA) was added to the cells and the cell suspension was gently mixed. Then, the cell suspension was incubated for 30 min at 37 °C, stirring every 10 min. After the end of the incubation, the cell suspension was vortexed for 60 s and centrifuged for 10 min at 500 rpm at room temperature. The resulting supernatant was transferred to a new centrifuge tube and centrifuged for 10 min at 700 rpm. The resulting supernatant was transferred to a new centrifuge tube and centrifuged for 15 min at 12,000 rpm. After centrifugation, the supernatant was removed, and the resulting CIMV pellet was dissolved in culture medium, PBS, or another buffer, depending on the purpose of further experiment. The concentration of total protein in the isolated CIMVs was assessed using a Pierce BCA Protein Assay Kit (Thermo Scientific, Waltham, MA, USA) according to the manufacturer’s recommended method.

### 2.6. Quantitative Polymerase Chain Reaction (qPCR)

To confirm the presence of IL2 mRNA, total RNA (tRNA) was extracted from hADSCs or CIMVs using a TRIzol Reagent (Invitrogen, USA) following the manufacturer’s instructions. Then, hADSC and CIMV tRNA were used for reverse transcription reaction to obtain cDNA using a GoScript™ Reverse Transcription System (Promega, Madison, WI, USA) according to manufacturer’s instructions. 

Primers and probes specific to 18S ribosomal RNA (18S rRNA) and IL2 cDNAs were designed using the GenScript Online Real-time PCR (TaqMan) Primer Design Tool (GenScript, Piscataway, NJ, USA) and synthesized by Evrogen (Moscow, Russia) (Table 1). TaqMan-based qPCR was performed in 96-well non-skirted PCR plates (#PCR-NH96, GenFollower, Shaoxing, China). The reaction mixture contained 1 μL of cDNA template, 0.3 μL of primers and a probe mix (final primer concentration of 300 nM each), 4.7 μL of sterile H_2_O (Evrogen, Moscow, Russia) and 4 μL of 10× TaqMan-buffer (Lytech, Moscow, Russia), in a final volume of 10 μL. The qPCR was carried out in a CFX96 Touch™ Real-Time PCR Detection System (BioRad Laboratories, Hercules, CA, USA) following the protocol: one cycle of denaturation at 95 °C for 3 min followed by 45 cycles of amplification at: 95 °C for 10 s, 55 °C for 30 s. The IL2 gene expression level was normalized to 18S rRNA level. Relative quantification was performed by the comparative threshold cycle (ΔΔCT) method.

### 2.7. Western Blot Analysis

To analyze the IL2 protein presence in hADSCs and CIMVs, 1 × 10^6^ cells or isolated from them CIMV were lysed in a RIPA buffer containing a Halt™ Protease and Phosphatase Inhibitor Cocktail (Thermo Scientific, Waltham, MA, USA). Equal volumes of protein samples (40 µg for hADSCs and 50 µg for CIMVs) were separated by electrophoresis in 4–12% SDS-PAGE gradient gels and transferred onto 0.2 μm nitrocellulose membranes (#162-0112, BioRad Laboratories, Hercules, CA, USA) using the Semi-dry electrophoretic transfer cell (BioRad Laboratories, Hercules, CA, USA). Membranes were blocked with 5% non-fat milk (#A0830, Applichem, Darmstadt, Germany) and then stained overnight at 4 °C with rabbit polyclonal anti-IL2 primary antibodies (1:2000; #ab180780, Abcam, Cambridge, UK). The removal of excess primary antibody was carried out by washing the membranes in Tris-buffered saline with 0.1% Tween-20 (TBST) four times for 15 min each. Then, the membranes were stained with HRP-conjugated goat anti-rabbit immunoglobulin G (1:2000; #8a0467j, American Qualex Antibodies, San Clemente, CA, USA) for 2 h at RT or with anti-β-actin horseradish peroxidase (HRP)-conjugated antibody (1:1000; #A00730, GenScript, Piscataway, NJ, USA) for 2 h at RT. Excessive secondary antibodies were removed by washing in TBST four times for 15 min each. Membranes were exposed to Clarity-enhanced chemiluminescence (ECL) reagent (#1705061, Bio-Rad Laboratories, Hercules, CA, USA) for 3 min at room temperature and visualized using a ChemiDoc XRS^+^ system (BioRad Laboratories, Hercules, CA, USA).

### 2.8. Immunophenotyping

To analyze the expression of cluster of differentiation (CD) markers typical for MSCs on the surface of hADSCs and isolate from them CIMVs, the following antibodies were used: CD90 (FITC) (#555595, BD Biosciences, San Jose, CA, USA), CD29 (APC) (#303008, BioLegend, San Diego, CA, USA), CD166 (PE) (#343904, BioLegend, San Diego, CA, USA), CD44 (PE) (#103024, BioLegend, San Diego, CA, USA), CD73 (APC) (#344006, BioLegend, San Diego, CA, USA), and a negative control (PE) (BD Stemflow™ Human MSC Analysis Kit, BD Biosciences, San Jose, CA, USA) including antibodies to CD34, CD11b, CD19, CD45, and HLA-DR. Briefly, 1 × 10^5^ native hADSCs, hADSCs-BFP and hADSCs-IL2 were trypsinized and washed two times with PBS, and stained with the antibodies for 30 min in the dark at room temperature. Cells were washed once with PBS and analyzed by flow cytometry using FACSAria III (BD Biosciences, San Jose, CA, USA), and data were analyzed using BD FACSDiva™ software version 7.0.

CIMVs were isolated from 2 × 10^5^ native hADSCs, hADSCs-BFP and hADSCs-IL2, washed once with PBS and stained with the listed above antibodies for 30 min in the dark at room temperature. Then, CIMVs were washed once with PBS and analyzed using the FACSAria III flow cytometer (BD Biosciences, San Jose, CA, USA). Data were analyzed using BD FACSDiva™ software version 7.0.

### 2.9. Apoptosis and Necrosis Detection

The percentages of apoptotic and necrotic cells were determined using an APC Annexin V Apoptosis Detection Kit with PI (#40932, BioLegend, San Diego, CA, USA) according to the manufacturer’s protocol. Native hADSCs, hADSCs-BFP and hADSCs-IL2 were seeded at a density of 5 × 10^4^ cells in 6-well plates. Cells were cultured in a DMEM/F12 medium for 48 h. Cells were conjugated with antibodies and analyzed by flow cytometry using the FACSAria III (BD Biosciences, San Jose, CA, USA). A minimum of 15,000 events were acquired for each sample. Two biological replicates were completed for the experiment.

### 2.10. MTS Cell Proliferation Assays

To analyze hADSC proliferation, 5 × 10^3^ native hADSCs, hADSCs-BFP and hADSCs-IL2 were plated in 100 μL of DMEM/F12 per well of a 96-well culture plate and incubated for 24 and 48 h. The proliferation of the hADSCs was evaluated using the CellTiter 96^®^ AQueous Non-Radioactive Cell Proliferation Assay kit (Promega, Madison, WI, USA) following the manufacturer’s instructions. The absorbance (490 nm) in the wells was measured using an Infinite M200Pro (Tecan Trading AG, Mannedorf, Switzerland) with reference wavelengths = 630 nm. Two biological replicates were completed for the experiment.

### 2.11. Determination of CIMV Size

The size of the CIMVs was determined by flow cytometry (BD FACSAria III, BD Biosciences, San Jose, CA, USA) with using a mixture of calibration particles (0.22–0.45–0.88–1.34 μm) (Spherotech, Lake Forest, IL, USA). In all the experiments with the CIMVs, a violet laser (Ex = 405, Em = 450) was used to detect particles from 200 nm in diameter. A minimum of 50,000 events were acquired for each sample. For sizing the CIMVs by scanning electron microscopy (SEM) analysis, CIMVs were isolated as previously described. Isolated CIMVs were resuspended in PBS and applied on glass slides by centrifugation at 3000 rpm for 30 min at room temperature. The CIMVs were fixed with 10% formalin for 15 min, dehydrated through an ethanol gradient from 30% to absolute and air-dried for 24 h. Prior to imaging, samples were coated with gold/palladium in a Quorum T150ES sputter coater (Quorum Technologies Ltd., Laughton, UK) and viewed for analysis by an SEM Merlin (Carl Zeiss, Jena, Germany). Two biological replicates were completed for the experiment.

### 2.12. Assessment of the Nuclear and Mitochondrial Components in CIMVs

To analyze the presence of nuclear and mitochondrial components in CIMVs, native hADSCs, hADSCs-BFP and hADSCs-IL2 were trypsinized and washed twice with PBS. Then, 5 × 10^5^ cells were resuspended in 1 mL of PBS and stained with 1 μL of DiD vital dye (Vybrant Multicolor Cell-Labeling Kit, Invitrogen, Waltham, MA, USA) for 10 min at room temperature in the absence of light. Then, cells were washed twice with PBS and a cell pellet was dissolved in 500 μL of PBS and stained with 2 μL of 300 nM MitoTracker Green FM (Molecular probes, Invitrogen, Waltham, MA, USA) for 15 min at 37 °C. After that, cells were washed twice with PBS and the pellet was dissolved in 500 μL of PBS and stained with 1.5 μL of the Hoechst 33,342 vital fluorescent dye (0.1 μg/mL) (Thermo Scientific, Waltham, MA, USA) for 10 min at room temperature in the absence of light. The cells were then washed and the CIMVs were isolated from them as previously described. The isolated CIMVs were analyzed on the CytoFLEX S flow cytometer (Beckman Coulter, Brea, CA, USA). A minimum of 50,000 events were acquired for each sample. Two biological replicates were completed for the experiment.

### 2.13. Cytokine Multiplex Analysis

The Human Chemokine 40-plex Panel (#171ak99mr2, BioRad Laboratories, Hercules, CA, USA) was used to analyze the conditioned medium, harvested from 2 × 10^6^ hADSCs at 24-, 48- and 72-h time points and centrifuged (1500 rpm for 10 min at room temperature), according to the manufacturer’s recommendations. Fifty microliters of each sample were used to determine cytokine concentration and the collected data were analyzed using a Luminex 200 analyzer with MasterPlex CT control and QT analysis software (MiraiBio division of Hitachi Software, San Francisco, CA, USA). Two biological replicates were completed for the experiment.

### 2.14. IL2 ELISA

The concentration of human IL2 protein in the lysates of hADSCs or CIMVs was determined using the ELISA MAX™ Deluxe Set Human IL-2 (#431804, BioLegend, San Diego, CA, USA) as recommended by the manufacturer. The hADSC protein lysates were prepared as previously described (Western blot analysis subsection). The CIMVs were isolated as previously described and resuspended in a RIPA buffer. The isolated proteins were used for the analysis. The IL2 concentration was calculated as pg/µg of total protein. The analysis was conducted in triplicate for two biological replicates.

### 2.15. Immunofluorescence Analysis of CIMVs

To analyze the morphology and the presence of the IL2 protein in native CIMVs and CIMVs-IL2, an immunofluorescence assay was carried out. For this, 1 × 10^6^ native hADSCs and hADSCs-IL2 were labeled with DiO (green) dye using a Vybrant Multicolor Cell-Labeling Kit (#V-22889, Invitrogen, Waltham, MA, USA) for general cell membrane labeling, according to the manufacturer’s instructions. After that, the CIMVs were isolated from the cells as previously described and pelleted on coverslips by centrifugation at 3200 rpm for 30 min at room temperature. The pelleted CIMVs were fixed with 250 μL of ice-cold methanol for 10 min at RT. The fixed CIMVs were washed 2 times for 5 min each in Tris-buffered saline (TBS; 50 mm Tris, 150 mm NaCl, pH 7.6); the CIMVs were centrifuged at 3200 rpm for 30 min after every wash step. The CIMVs were then incubated with primary anti-IL2 antibodies (ab180780, Abcam, Cambridge, UK 1:100 dilution in TBS) for 1 h at room temperature. After that, the CIMVs were washed 2 times for 5 min in TBS and then incubated with secondary antibodies (goat anti-rabbit IgG (H+L) Alexa Fluor 647, Invitrogen, USA, 1:100 dilution in TBS) for 1 h at room temperature, and washed again. Coverslips were mounted on the slides with a mounting medium (ImmunoHistoMount, Santa Cruz Biotechnology, Santa Cruz, CA, USA). The samples were investigated under an LSM 780 confocal microscope (Carl Zeiss, Jena, Germany) using Zen black 2012 software (Carl Zeiss, Jena, Germany) using identical confocal settings (laser intensity, gain, and offset).

### 2.16. Immune Cell Activation

In order to analyze the immunomodulatory activity of hADSCs and CIMVs isolated from native and genetically modified hADSCs, PBMCs were isolated from 4 different donors as previously described [31]. To analyze the effect of cells on PBMC activation, native hADSCs, hADSCs-BFP and hADSCs-IL2 were seeded (1 × 10^5^ cells per well) in 12-well plates and incubated for 24 h. After 24 h, 2 × 10^6^ PBMCs per well in 2 mL of IMDM were added to hADSCs and they were incubated with PBMCs for 72 h.

In order to analyze the effect of CIMVs on PBMC activation, PBMCs were seeded in a number of 2 × 10^6^ cells per 3 cm ultra-low attachment culture dish, then native CIMVs, CIMVs-BFP and CIMVs-IL2 (50 μg/mL in 2 mL of IMDM) were added to PBMCs. Cells were incubated with CIMVs for 72 h.

The activation of PBMC populations was determined by flow cytometry using staining with antibodies specific to surface markers of various human immune cell populations. After 72 h, PBMCs treated with hADSCs or CIMVs were washed twice with PBS and stained with the following antibodies for 30 min in the absence of light at room temperature: PE anti-human HLA-DR antibody (#307605, BioLegend, San Diego, CA, USA), APC anti-human CD25 antibody (#302610, BioLegend, San Diego, CA, USA), FITC anti-human CD8a antibody (#300906, BioLegend, San Diego, CA, USA), PE anti-human CD3 antibody (#300308, BioLegend, San Diego, CA, USA), Pacific Blue anti-human CD4 antibody (#317429, BioLegend, San Diego, CA, USA), PerCP/Cy5.5 anti-human CD38 antibody (#356614, BioLegend, San Diego, CA, USA), PE anti-human CD127 (IL-7Rα) antibody (#351304, BioLegend, San Diego, CA, USA), APC anti-human CD183 (CXCR3) antibody (#353708, BioLegend, San Diego, CA, USA), PE anti-human CD196 (CCR6) antibody (#353410, BioLegend, San Diego, CA, USA), APC anti-human CD4 antibody (#300514, BioLegend, San Diego, CA, USA), FITC anti-human CD3 antibody (#317306, BioLegend, San Diego, CA, USA), PerCP/Cyanine5.5 anti-human CD56 (NCAM) antibody (#362506, BioLegend, San Diego, CA, USA). Then, the cells were washed twice with PBS and analyzed on the FACSAria III flow cytometer (BD Biosciences, San Jose, CA, USA) and data were analyzed using BD FACSDiva™ software version 7.0. A minimum of 20,000 events were acquired for each sample. Flow cytometry results were obtained in 1 replicate for 4 different donors and summarized together during analysis (*n* = 4).

### 2.17. T-Cell Proliferation Assay

In order to analyze the effect on PBMC proliferation, native hADSCs, hADSCs-BFP and hADSCs-IL2 were seeded (5 × 10^4^ cells per well) in 24-well plates and incubated for 24 h. PBMCs were isolated as previously described and labeled with 1 μM of 5,6-carboxyfluorescein succinimidyl ester (CFDA) (eBioscience, Thermo Scientific, Waltham, MA, USA) fluorescent dye for 10 min in the absence of light at room temperature. PBMCs (1 × 10^6^ PBMCs) were added into each well and stimulated with Phytohemagglutinin-M (10 μg/mL; PHA) (PanEco, Moscow, Russia) or a combination of CD3 and CD28 antibodies (0.1 µg/mL each) (both GenScript, Piscataway, NJ, USA) for 72 h at 37 °C with 5% CO_2_.

In order to analyze the effect of CIMVs on PBMC proliferation, CFDA-labeled PBMCs were plated in a number of 1 × 10^6^ cells per well in 4-well ultra-low attachment plates, and after that native CIMVs, CIMVs-BFP and CIMVs-IL2 in concentration 25 µg/mL and PHA (10 μg/mL) or CD3+CD28 (0.1 µg/mL each) antibodies in 1 mL of IMDM were added to PBMCs. Cells were incubated with CIMVs for 72 h.

At the end of the incubation, PBMCs were collected and labeled with anti-CD4 and ani-CD8a antibodies (PE, #300508 and APC, #300912, respectively, BioLegend, San Diego, CA, USA), and CFDA fluorescence was analyzed using a CytoFLEX S with CytExpert software version 1.2 (Beckman Coulter, Brea, CA, USA), a minimum of 20,000 events were acquired for each sample.

T-cell proliferation was calculated as previously described [32]. The negative control, in which PBMCs remained unstimulated (no PHA or CD3+CD28 was added), was used to define the threshold of CFDA signal for non-proliferating T-cells. The number of cells with lower CFDA per cell (as compared to the negative control) was accepted as the number of proliferating cells. Bars represent the mean ± SD (*n* = 4) of four biological replicates (one technical replicate for each donor).

### 2.18. Analysis of Activated PBMC Cytotoxicity on Human Triple Negative Breast Cancer Cells

In order to analyze the anti-tumor activity of PBMCs after interaction with hADSCs or CIMVs, PBMCs were activated for 72 h as described in previous section. MDA-MB-231 tumor cells were seeded on 16-well xCelligence plates (3 × 10^3^ tumor cells per well). MDA-MB-231 or MDA-MB-436 were seeded on 6-well plates (5 × 10^4^ tumor cells per well). After that, 72 h-activated PBMCs were added to MDA-MB-231 or MDA-MB-436 tumor cells in a 1:3 ratio (tumor cells:PBMCs). The proliferative activity of MDA-MB-231 tumor cells was assessed using a RTCA xCelligence device (ACEA Biosciences, San Diego, CA, USA) for 72 h. The viability of MDA-MB-231 or MDA-MB-436 tumor cells was assessed 24 h after the addition of PBMCs using an APC Annexin V Apoptosis Detection Kit with PI (#40932, BioLegend, San Diego, CA, USA). Stained cells were analyzed on the FACSAria III flow cytometer (BD Biosciences, San Jose, CA, USA) and data were analyzed using BD FACSDiva™ software version 7.0. The obtained results are calculated as the sum of four biological replicates (one technical replicate for each PBMC type isolated from four different donors).

### 2.19. Animals

To analyze the effect of CIMVs-IL2 on the immune cells in vivo, 8-week-old male mice (CD-1 strain) were used. The animals were housed in transparent plastic cages, 12:12 h light:dark cycle, at a temperature between 21–25 °C, with a relative humidity of 40% to 70% with ad libitum food and water access. All experiments were carried out in compliance with the procedure protocols approved by the Kazan Federal University local ethics committee (protocol No. 3, date 03/23/2017) according to the rules adopted by Kazan Federal University and Russian Federation Laws. The mice were intravenously injected with 200 µL of PBS, 50 µg (in 200 µL of PBS) of native CIMVs, CIMVs-BFP or CIMVs-IL2. Each group contained five animals. Mice were euthanized in compliance with the procedure protocols approved by the Kazan Federal University local ethics committee (protocol No. 3, date 03/23/2017). The peripheral blood of mice was collected in 1.5 mL tubes with heparin, and mononuclear blood cells were immediately isolated.

### 2.20. Isolation of Murine Peripheral Blood Mononuclear Cells

Murine PBMCs were isolated using a Histopaque-1077 (Sigma-Aldrich, St. Louis, MO, USA) gradient as previously described in [33]. Collected murine blood was diluted with PBS (1:1) and 1 volume of blood was layered onto 1 volume of the Histopaque-1077. The blood and Histopaque-1077 were centrifuged at 400× *g* for 30 min at room temperature. After separation, the opaque interface containing PBMCs was collected into new tube. The PBMCs were washed twice with PBS and used for the following analysis.

### 2.21. Analysis of Murine PBMC Populations

To analyse the effect of CIMVs-IL2 on the number of murine PBMCs, newly isolated cells were stained with antibodies specific for the surface markers typical for various populations of mouse immune cells. Briefly, 5 × 10^5^ murine PBMCs in 100 µL of PBS were stained with the following antibodies for 30 min in the absence of light at RT: CD45-PerCP (#103130; BioLegend, San Diego, CA, USA), CD3-PE (#100308; BioLegend, San Diego, CA, USA), CD8a-PE/Cy7 (#100722; BioLegend, San Diego, CA, USA), CD4-APC (#100412; BioLegend, San Diego, CA, USA). After staining, the cells were washed twice with PBS and analyzed by flow cytometry using the FACSAria III (BD Biosciences, San Jose, CA, USA) and data were analyzed using BD FACSDiva™ software version 7.0. A minimum of 20,000 events were acquired for each sample. Flow cytometry results were obtained in one replicate for each animal and summarized together (*n* = 5, five biological replicates).

### 2.22. Multiplex Analysis of Mouse Cytokines

To analyse changes in the cytokine profile of mice after the injection of native CIMVs, CIMVs-BFP or CIMVs-IL2, the Bio-Plex Pro Mouse Cytokine 23-plex panel (#M60009RDPD, BioRad Laboratories, Hercules, CA, USA) was used. Serum was collected from mice 72 h after the injection of CIMVs and used for the analysis according to the manufacturer’s recommendations. Fifty microliters of each serum sample for the analysis and the data were analyzed using a Luminex 200 analyzer with MasterPlex CT control and QT analysis software (MiraiBio division of Hitachi Software, San Francisco, CA, USA). Five biological replicates were completed for the experiment.

### 2.23. Statistical Analysis

Statistical analysis was achieved using GraphPad Prism 7 software (GraphPad Software, San Diego, CA, USA), one-way ANOVA followed by Tukey HSD post-hoc comparisons test. Significant probability values are denoted as * *p* < 0.05, ** *p* < 0.01 and *** *p* < 0.001, **** *p* < 0.0001.

## 3. Results

### 3.1. IL2 Expression Failed to Affect the Viability, Proliferative Activity and Ability to Direct Differentiation of hADSCs

Mesenchymal stem cells were isolated from human adipose tissue. Cell viability following isolation was determined as 95% by trypan blue staining. hADSCs had a fibroblast-like morphology and a pattern of surface antigen expression specific for human MSCs, positive for CD29, CD44, CD73, CD90 and CD166, and negative for CD34, CD11b, CD19, CD45, and HLA-DR, the latter being markers typical for hematopoietic cells (Appendix A).

Genetically modified hADSCs expressing IL2 or BFP were produced using lentiviral transduction. The expression of IL2 in hADSCs-IL2 was confirmed using qPCR and Western blot. BFP expression was detected by fluorescence microscopy. Increased levels of IL2 gene transcripts (*n* = 6, *p* < 0.0001) were demonstrated in hADSCs-IL2 compared to native hADSCs and hADSCs-BFP (Figure 1a). IL2 protein secretion was analyzed in conditioned media (CM) harvested from the native hADSCs, hADSCs-BFP or hADSCs-IL2, and was found to be significantly higher in hADSCs-IL2 (2255.7 ± 738.3 pg/mL, *p* < 0.0001) compared to native hADSCs (2.5 ± 0.2 pg/mL) and hADSCs-BFP (2.0 ± 0.06 pg/mL) at 24 h (Figure 1b). This significant increase in IL2 secretion by hADSCs-IL2 was maintained when analyzed in CM at 48 (hADSCs-IL2 – 1602.83 ± 68.9 pg/mL; native hADSCs—2.03 ± 0.31 pg/mL; hADSCs-BFP—1.758 ± 0.16 pg/mL) and 72 h (hADSCs-IL2 – 4322.86 ± 504.64 pg/mL; native hADSCs—2.51 ± 0.0 pg/mL; hADSCs-BFP—2.34 ± 0.24 pg/mL) (Figure 1b). Total cell IL2 expression (non-secreted) was also demonstrated using Western blot analysis (Figure 1c). The concentration of IL2 in hADSCs-IL2 was 10.56 ± 0,71 pg/µg of total protein.

The proliferative activity of native hADSCs, hADSCs-BFP and hADSCs-IL2 was determined at 24 and 48 h post seeding in vitro, and no significant differences were determined (Figure 1d). The number of apoptotic and necrotic cells were also determined at 24 and 48 h post seeding in vitro as a marker of cell viability. At 24 h, no significant difference in the number of apoptotic or necrotic cells were found in native or modified hADSCs. However, at 48 h, the viability of hADSCs-IL2 increased slightly with more non-apoptotic/necrotic cells detected (91.6% ± 1.3%, *p* < 0.05) compared to native hADSCs (83.0% ± 1.2%) and hADSCs-BFP (83.6% ± 1.1%) (Figure 2).

Like native hADSCs, the genetically modified hADSCs-IL2 were able to differentiate into adipogenic, osteogenic and chondrogenic lineages (Figure 3). Together, these data indicate that hADSCs-IL2 express IL2 mRNA and protein which did not affect the proliferative properties and viability, and the differentiation properties of this cell population.

### 3.2. CIMVs Isolated from hADSCs-IL2 Carry IL2 mRNA and Protein

Artificial CIMVs were isolated from native and genetically modified hADSCs following treatment with Cytochalasin B. It was shown that the size of the isolated CIMVs was predominantly 220 nm or below (66.9 ± 3.8% of native CIMVs, 72.5 ± 2.7% of CIMVs-BFP and 71.8 ± 3.4% of CIMVs-IL2) (Figure 4a). Moreover, the population of isolated CIMVs also contained a proportion of vesicles ranging in size from 220 to 450 nm (17.0 ± 1.8% of native CIMVs, 15.5 ± 0.4% of CIMVs-BFP and 15.2 ± 1.4% of CIMVs-IL2), as well as much larger vesicles ranging in size from 450 to 880 nm (11.9 ± 1.9% of native CIMVs, 11.0 ± 1.9% of CIMVs-BFP and 11.6 ± 2.0% of CIMVs-IL2) (Figure 4a). A small number of CIMVs ranging in size from 880 nm to 1340 nm were also observed, albeit at very low concentrations (1.1 ± 0.2% of native CIMVs, 1.0 ± 0.4% of CIMVs-BFP and 1.3 ± 0.1% of CIMVs-IL2) (Figure 4a). The morphology and size of the CIMVs was analyzed by SEM. CIMVs had the characteristic spherical shape and varied in size reflecting the previous characterization data. Some CIMV deformation was observed; this is a common occurrence when preparing vesicles samples for SEM analysis and is not an artifact of the Cytochalasin B production process (Figure 4b).

Analysis of the presence of nuclear and mitochondrial components showed the presence of mitochondria in 23.78 ± 3.7% of CIMVs isolated from native hADSCs, 21.42 ± 1.56% of CIMVs-BFP and 21.36 ± 2.89% of CIMVs-IL2 (Figure 4c). The fluorescence of a nuclear component was detected in 7.6 ± 0.9% of CIMVs isolated from native hADSCs, in 5.6 ± 1.5% of CIMVs-BFP, in 6.1 ± 2.9% of CIMVs-IL2 (Figure 4d). The number of native CIMVs containing both nuclear and mitochondrial components was 4.37 ± 1.24%, the number of CIMVs-BFP was 3.41 ± 0.62% and the number of CIMVs-IL2 was 3.17 ± 0.58% (Figure 4e).

The presence of IL2 mRNA and IL2 protein in CIMVs was analyzed. Using qPCR, it was shown that CIMVs-IL2 contained by 493.16 ± 1.0 times more IL2 mRNA in comparison with native CIMVs and CIMVs-BFP (Figure 4f). The presence of IL2 protein (17 kDa) was also shown in CIMVs-IL2, in contrast to native CIMVs and CIMVs-BFP using Western blot (Figure 4g) and immunofluorescence assay (Appendix A). The concentration of IL2 in CIMVs-IL2 was 9.98 ± 0.32 pg/µg of total protein.

The received data indicate that isolated CIMVs-IL2 contain both IL2 mRNA and protein and the genetic modification fails to affect the size and the number of mitochondrial and nuclear components in CIMVs.

### 3.3. CIMVs Carry a Reduced Number of Parental Membrane CD Markers

In order to determine whether CIMVs carry receptors typical for the parental cells on their surface, immunophenotyping of both parental cells and CIMVs isolated from them was carried out. There was no statistically significant difference between native hADSCs, hADSCs-BFP and hADSCs-IL2. In all hADSC cultures, the expression of CD29, CD44, CD73, CD90, CD166 markers was about 98%, 99%, 90%, 80% and 50% of cells, respectively (Figure 5). As expected, the expression of common MSC surface markers was significantly reduced in both native CIMVs and CIMVs-BFP and CIMVs-IL2. CIMVs expressed parental markers as follows: CD29 25%, CD44 70% cells, CD73 52%, CD90 80%, and CD166 25% (Figure 5).

### 3.4. Both hADSCs-IL2 and CIMVs-IL2 Can Activate Human PBMCs

To evaluate the effect of hADSCs-IL2 and CIMVs-IL2 on the activation of human PBMCs, newly isolated PBMCs from four different donors were incubated with 1 × 10^5^ (which is equivalent to 50 µg of total protein) of native or genetically modified hADSCs, or with 50 µg of CIMVs isolated from native or genetically modified hADSCs for 72 h. The gating strategy used to distinguish the various lymphocyte populations is depicted in Figure 6a. The number of activated T-killers (CD3^+^ CD8^+^ CD4^−^ CD38^+^ HLA-DR^+^) was significantly increased (*n* = 4, *p* < 0.0001) after incubation with hADSCs-IL2 (179.1 ± 5.2%) compared to control PBMCs incubated separately (100.0 ± 1.5%). An increase was also found in PBMCs incubated with native hADSCs (149.8 ± 0.0%) and hADSCs-BFP (130.6 ± 2.5%), but it was significantly lower than in the sample of PBMCs incubated with hADSCs-IL2 (*n* = 4, *p* < 0.0001) (Figure 6b). The incubation of PBMCs with CIMVs-IL2 also led to a significant increase (*n* = 4, *p* < 0.0001) in the number of CD3^+^ CD8^+^ CD4^−^ CD38^+^ HLA-DR^+^ T-killers (205.6 ± 4.5%) in comparison with native PBMCs (100.0 ± 9.3%) and PBMCs incubated with native CIMVs (116.7 ± 4.5%) and CIMVs-BFP (113.2 ± 0.8%) (Figure 6b). 

An increase in the number of CD3^−^CD56^+^ NK cells (*n* = 4, *p* < 0.0001) was also observed in the sample of hADSCs-IL2 (122.9 ± 3.6%) compared to control PBMCs, incubated alone (100.0 ± 0.4%) or with native hADSCs (95.4 ± 2.2%). However, the hADSCs-BFP sample also showed an increase (124.9 ± 0.9%), which was not statistically different from the hADSCs-IL2 sample (Figure 6b). In all cases where PBMCs were incubated with vesicles, the number of NK cells was decreased (native CIMVs—78.2 ± 2.2%, CIMVs-BFP—64.1 ± 2.2%, CIMVs-IL2—71.7 ± 5.8%) (*n* = 4, *p* < 0.01) compared to control PBMCs (100.0 ± 10.1%) (Figure 6b). 

The population of Th1 cells (CD3^+^CD4^+^CD8^−^CCR6^−^CXCR3^+^) was also increased in all PBMCs after incubation with hADSCs (native hADSCs: 131.5 ± 1.6%, hADSCs-BFP: 126.0 ± 3.4%, hADSCs-IL2: 114.3 ± 1.3%), compared to controls (100.0 ± 2.9%). However, the increase in hADSCs-IL2 was the lowest (*n* = 4, *p* < 0.01) (Figure 6b). After incubation with CIMVs, the number of Th1 cells did not change (Figure 6b). Similarly, the number of Th2 (CD3^+^CD4^+^CD8^−^CCR6^−^CXCR3^−^) was slightly increased (*n* = 4, *p* < 0.01) in the PBMCs cultivated with native hADSCs (116.0 ± 2.1%). However, there were no significant changes when PBMCs were incubated with the genetically modified hADSCs (hADSCs-BFP: 99.3 ± 3.0%, hADSCs-IL2: 98.6 ± 5.4%, control PBMCs: 100.0 ± 1.1%) (Figure 6b). CIMVs had almost no effect on the Th2 population (PBMCs: 100.0 ± 0.8%, native CIMVs: 103.5 ± 0.5%, CIMVs-BFP: 94.7 ± 0.8%), with a small increase (*n* = 4, *p* < 0.0001) found after the application of CIMVs-IL2 e (109.5 ± 0.1%) (Figure 6b). 

The incubation of PBMCs with genetically modified hADSCs-BFP (150.0 ± 8.8%) and hADSCs-IL2 (145.8 ± 2.9%) resulted in an increase in the number of Th17 cells (CD3^+^CD4^+^CD8^−^CCR6^+^CXCR3^−^) (*n* = 4, *p* < 0.0001) in comparison with native hADSCs (110.4 ± 5.8%) and PBMCs cultured alone (100.0 ± 0.0%) (Figure 6b). While co-cultivation with CIMVs isolated from genetically modified hADSCs (CIMVs-BFP: 83.3 ± 5.7%, CIMVs-IL2: 66.6 ± 5.7%) led to a decrease in the number of Th17 cells (*n* = 3, *p* < 0.01) compared to native CIMVs (103.3 ± 0.5%) and control PBMCs (100.0 ± 10.0%) (Figure 6b). Similarly, the number of Tregs (CD3^+^CD4^+^CD8^−^CD25^+^CD127^low^) was increased (*n* = 4, *p* < 0.01) after incubation with native hADSCs (128.8 ± 4.1%), as well as with hADSCs-BFP (119.2 ± 1.0%) and hADSCs-IL2 (120.7 ± 7.3%) compared to PBMCs alone (100.0 ± 1.0%) (Figure 6b). Whilst incubation with CIMVs from all hADSCs lines failed to change the number of Tregs (PBMCs: 100.0 ± 11.5%, native CIMVs: 106.9 ± 2.7%, CIMVs-BFP: 97.5 ± 2.0%, CIMVs-IL2: 127.2 ± 19.5%) (Figure 6b).

In summary, both CIMVs-IL2 can increase the number of activated CD8^+^ T-killers more effectively than hADSCs-IL2.

### 3.5. CIMVs Had Failed to Influence T-Cell Proliferation

In order to evaluate the effect of hADSCs and CIMVs on the proliferative activity of T-cells, the proliferation of PBMCs isolated from peripheral blood was stimulated with PHA or with a CD3 + CD28 antibody combination to mimic antigen presentation. Analysis of PHA-stimulated T-cell proliferation after co-culture with hADSCs showed that co-cultivation with native hADSCs (32.24 ± 1.03% of proliferated cells) as well as hADSCs-BFP (45.28 ± 0.12%) and hADSCs-IL2 (40.37 ± 0.68%) had a significant suppressive effect on the proliferation of CD4^+^ T-cells compared to T-cells stimulated with PHA only (77.58 ± 0.5%) after 72 h (*n* = 4, *p* < 0.0001) (Figure 7). At the same time, proliferation of PHA-stimulated CD8^+^ T-cells was also decreased after co-culture with hADCSs-IL2 (23.96 ± 0.72%), hADSCs-BFP (31.14 ± 0.88%) and with native hADSCs (25.84 ± 0.62%) compared to PHA-only stimulated cells (56.84 ± 0.47%) after 72 h (*n* = 4, *p* < 0.0001) (Figure 7). Whilst the proliferation of PHA-stimulated CD4^+^ T-cells remained unchanged after co-culture with native CIMVs (75.35 ± 0.41%), CIMVs-BFP (76.15 ± 0.82%) and CIMVs-IL2 (75.69 ± 0.48%) compared to PHA-only stimulated CD4^+^ T-cells (74.43 ± 0.89%) (Figure 7). Similar results were obtained for PHA-stimulated CD8^+^ T-cell where proliferation after cultivation with CIMVs-BFP (62.99 ± 0.45%) and CIMVs-IL2 (61.36 ± 0.50%) was the same compared to PHA-only stimulated CD8^+^ T-cells (59.75 ± 1.66%) (Figure 7). A slight decrease was observed in CD8^+^ T-cell proliferation co-cultured with native CIMVs (52.87 ± 2.53%) (Figure 7).

With the CD3 and CD28 antibodies, PBMC proliferation was significantly lower compared to PHA. The addition of CD3 + CD28 did not lead to a significant increase in the proliferation of CD4^+^ T-cells, neither when T-cells were cultured alone (1.15 ± 0.13%), nor after co-culture with native hADSCs (0.98 ± 0.02%) or hADSCs-BFP (1.6 ± 0.33%) compared with an unstimulated control (1.75 ± 0.72%). The number of CD4^+^ T-cells was increased only in hADSCs-IL2 sample (7.70 ± 0.1%) (*n* = 4, *p* < 0.0001) (Figure 7). The same increase in the number of CD8^+^ T-cells was observed in PBMCs cultured with hADSCs-IL2 (7.74 ± 0.05%) (*n* = 4, *p* < 0.0001), while the number of T-cells cultured alone (1.09 ± 0.16%), with native hADSCs (0.94 ± 0.07%) or hADSCs-BFP (1.56 ± 0.38%) remained unchanged compared to the unstimulated control (1.59 ± 0.50%) (Figure 7). The cultivation with CIMVs-IL2 resulted in a slight increase in the proliferation of CD4^+^ T-cells (3.94 ± 0.26%) (*n* = 4, *p* < 0.05), compared to unstimulated CD4^+^ T-cells (1.45 ± 0.72%), CD3/CD28-stimulated T-cells cultured alone (2.47 ± 0.30%), with native CIMVs (1.97 ± 0.16%) or CIMVs-BFP (1.97 ± 0.43%) (Figure 7). Whilst the number of CD8^+^ T-cells was increased in the samples cultured with CIMVs compared to unstimulated CD4^+^ T-cells (1.14 ± 0.24%), and CD3/CD28-stimulated T-cells cultured alone (1.62 ± 0.03%) (*n* = 4, *p* < 0.0001), there was no difference between native CIMVs (2.25 ± 0.38%), CIMVs-BFP (2.06 ± 0.02%) and CIMVs-IL2 (2.5 ± 0.23%) (Figure 7).

Together, CIMVs-IL2 do not suppress the proliferation of PHA-stimulated T-cells, in contrast to hADSCs-IL2. However, the ability of CIMVs-IL2 to stimulate cell proliferation after antigen presentation is much less noticeable that the same of hADSCs-IL2.

### 3.6. CIMV-IL2-Activated T-Cells Can Kill Human Triple Negative Breast Cancer Cells

In order to evaluate the cytotoxic activity of PBMCs activated by hADSCs or their CIMVs, MDA-MB-231 or MDA-MB-436 triple negative breast cancer cells were exposed to PBMCs activated by either the hADSCs lines or CIMVs to determine the effects on proliferative activity and the number of apoptotic/necrotic tumor cells. The cultivation of MDA-MB-231 cells with PBMCs activated by hADSCs-IL2 resulted in a significant decrease in the proliferation of cancer cells by 36% (to 63.82 ± 14.1%) (*n* = 4, *p* < 0.0001) after 24 h compared to untreated MDA-MB-231 (100.0 ± 4.8%) and MDA-MB-231 incubated with unactivated PBMCs (98.8 ± 8.3%). PBMCs activated by native hADSCs (93.0 ± 15.5%) or hADSCs-BFP (141.2 ± 0.1%) did not replicate this reduction in MDA-MB-231 proliferation (Figure 8a,c). After 72 h of cultivation with PBMCs activated with hADSCs-IL2, MDA-MB-231 proliferation was still decreased (74.16 ± 1.12%) (*n* = 4, *p* < 0.0001) in comparison with untreated MDA-MB-231 (100.00 ± 4.66%) and MDA-MB-231 incubated with unactivated PBMCs (87.16 ± 4.79%). However, the decrease was not as significant (by 25%) as after 24 h (Figure 8a,d). The number of nonapoptotic and non-necrotic MDA-MB-231 cells was also decreased after cultivation with hADSC-IL2-activated PBMCs (59.75 ± 0.07%) (*n* = 4, *p* < 0.0001) compared to untreated MDA-MB-231 (81.05 ± 0.92%) and MDA-MB-231 incubated with unactivated PBMCs (73.55 ± 0.07%). Again, this effect on cell apoptosis and necrosis was not replicated by PBMCs activated by native hADSCs (75.6 ± 0.05%) or hADSCs-BFP (65.0 ± 0.01%) (Figure 8b). 

After the addition of PBMCs activated with CIMVs-IL2, the proliferation of MDA-MB-231 cells was very significantly decreased. Twenty-four hours after the addition of CIMV-IL2-activated PBMCs, the proliferation of MDA-MB-231 was decreased by four-fold (to 25.0 ± 5.6%) compared to untreated MDA-MB-231 (100.0 ± 2.9%) and MDA-MB-231 incubated with unactivated PBMCs (110.8 ± 8.7%). As seen with the hADSCs, PBMCs activated by native CIMVs (103.7 ± 2.4%) of CIMVs-BFP (124.1 ± 58.3%) did not replicate this growth inhibitory effect (Figure 8e,g). The number of healthy MDA-MB-231 cells after 24 h of cultivation with PBMCs activated with CIMVs-IL2 was also reduced by 35% (46.1 ± 3.7%) compared to untreated MDA-MB-231 (78.2 ± 2.1%) and MDA-MB-231 incubated with unactivated PBMCs (70.1 ± 2.8%). Similarly, PBMCs activated by native CIMVs (62.1 ± 3.2%) or CIMVs-BFP (62.5 ± 3.4%) did not demonstrate the same effect on MDA-MB-231 viability (Figure 8f). The reduction in proliferation of MDA-MB-231 cells that remained viable after incubation with PBMCs activated by CIMVs-IL2 continued (41.2 ± 4.7%) after 72 h when compared to untreated MDA-MB-231 (100.0 ± 21.0%) and MDA-MB-231 incubated with unactivated PBMCs (109.7 ± 2.5%) (Figure 8e,h).

As a conclusion, CIMVs-IL2-activated T-killers are able to effectively kill human triple negative cancer cells MDA-MB-231. The cytotoxic activity of hADSCs-IL2 is less noticeable.

The cultivation of MDA-MB-436 cells with hADSCs-IL2-activated PBMCs for 24 h resulted in apoptosis induction in cancer cells (the number of nonapoptotic and non-necrotic cancer cells was 73.6 ± 2.8%, *n* = 4, *p* < 0.001) compared to untreated MDA-MB-231 (90.4 ± 1.4%), MDA-MB-231 incubated with unactivated PBMCs (83.9 ± 0.6%), native hADSCs-activated PBMCs (84.2 ± 1.6%) or hADSCs-BFP-activated PBMCs (83.1 ± 0.2%) (Figure 9a). MDA-MB-436 cells were also incubated with PBMCs which were activated with CIMVs. After that, the number of non-apoptotic and non-necrotic MDA-MB-436 cells was decreased after cultivation with PBMCs activated by CIMVs-IL2 by 28% (61.0 ± 2.7%, *n* = 4, *p* < 0.01) compared to untreated MDA-MB-231 (89.3 ± 1.9%) and MDA-MB-231 incubated with unactivated PBMCs (80.9 ± 0.9%), native hADSCs-activated PBMCs (75.8 ± 2.9%) or hADSCs-BFP-activated PBMCs (72.7 ± 4.6%) (Figure 9b).

### 3.7. CIMVs-IL2 Failed to Affect Murine CD8^+^ T-Cells but Affected Cytokine Profile

To analyze the effect of CIMVs-IL2 in in vivo, CD-1 immunocompetent mice were intravenously injected with 50 µg (in 200 µL of PBS) of native CIMVs, CIMVs-BFP or CIMVs-IL2. After 72 h, peripheral blood was collected and murine peripheral blood mononuclear cells were isolated and stained with antibodies typical for murine immune cell populations. There was no statistically significant difference in the number of murine CD45^+^CD3^+^CD4^−^CD8^+^ T-cells after cultivation with CIMVs-IL2 (number of CD8^+^ T-cells after treatment with PBS only 13.71 ± 2.99%, native CIMVs 11.47 ± 2.94%, CIMVs-BFP 11.85 ± 3.30%, and CIMVs-IL2 12.56 ± 1.69%) (Figure 10a). Cultivation with CIMVs also failed to affect the number of CD45^+^CD3^+^CD8^−^CD4^+^ T-cells (PBS only 28.98 ±11.31%, native CIMVs 23.17 ± 5.58%, CIMVs-BFP 21.65 ± 4.39%, and CIMVs-IL2 29.6 ± 2.79%) (Figure 10b).

Analysis of the cytokine profile of mouse serum after the administration of CIMVs showed that the amount of secreted IL3 was significantly increased after the injection of native CIMVs (15.6 ± 7.9 pg/mL, *p* < 0.01) and CIMVs-BFP (12.6 ± 7.2 pg/mL, *p* < 0.01) compared to the PBS-only injected group (2.8 ± 1.4 pg/mL). After the administration of CIMVs-IL2 (1.3 ± 1.5 pg/mL), the IL3 level was not significantly different from the IL3 level observed in the PBS group (Figure 11a). Similarly, the level of IL13 was also increased after the injection of native CIMVs (77.8 ± 20.0 pg/mL, *p* < 0.05) and CIMVs-BFP (91.4 ± 11.2 pg/mL, *p* < 0.01), whilst levels were not significantly different to the control PBS group (45.7 ± 22.6 pg/mL) after CIMVs-IL2 administration (24.8 ± 13.3 pg/mL) (Figure 11b). The same pattern was observed in the secretion of monocyte chemoattractant protein 1 (MCP-1), where the chemokine levels were higher after the administration of native CIMVs (325.1 ± 108.3 pg/mL, *p* < 0.05) and CIMVs-BFP (287.3 ± 41.7 pg/mL) in comparison with PBS (170.5 ± 94.3 pg/mL) and CIMVs-IL2 (54.3 ± 72.8 pg/mL) (Figure 11c). At the same time, the secretion of granulocyte colony-stimulating factor (G-CSF) was significantly reduced in samples of native CIMVs (22.1 ± 6.8 pg/mL, *p* < 0.001) and CIMVs-BFP (38.5 ± 4.9 pg/mL, *p* < 0.001) in contrast to the group of animals injected with PBS (765.0 ± 524.6 pg/mL) or CIMVs-IL2 (772.8 ± 54.6 pg/mL) (Figure 11d). The secretion of the chemokine eotaxin was significantly reduced after the administration of CIMVs-IL2 (90.7 ± 112.0 pg/mL, *p* < 0.05) compared to the PBS (490.6 ± 185.7 pg/mL), native CIMVs (360.5 ± 32.3 pg/mL) and CIMVs-BFP (309.2 ± 149.7 pg/mL) (Figure 11e). The secretion of cytokines IFN-γ, IL2, IL5, IL12p40 and macrophage inflammatory protein 1α (MIP-1α) also tended to increase in serum samples after the administration of native CIMVs or CIMVs-BFP, and to decrease in samples after the injection of CIMVs-IL2 (Table 2). However, the differences were not statistically significant. No significant changes or trends were observed in the levels of the other cytokines in the panel (Table 2).

## 4. Discussion

MSCs offer a potential advance for cell-mediated cancer therapy, as they can serve as effective vehicles for therapeutic delivery. MSCs expressing immunomodulating molecules can enhance the anti-tumor immune response. To determine to therapeutic effect of MSCs, we have generated and characterized a genetically engineered hADSCs-IL2 overexpressed IL2. The expression of human IL2 in hADSCs had no effect on the ability to direct differentiation into adipogenic, osteogenic and chondrogenic directions of the hADSCs or on their proliferative capacity, although a modest decrease in the number of apoptotic/necrotic hADSCs-IL2 compared to native hADSCs and hADSCs-BFP was detected. This may in part be due to autocrine and/or paracrine stimulation by IL2 molecules secreted by modified hADSCs-IL2, since IL2 can upregulate the cell cycle pathway in hADSCs and increase proliferation [34].

Almost all cells in the body secrete various types of EVs, which are an important mechanism of intercellular communication, serving as vehicles for the transfer of membrane and cytosolic proteins, lipids and nucleic acids between cells [35]. Since EVs carry parental molecules, they are proposed as promising tools for drug delivery in cancer therapy. However, it is difficult to obtain a sufficient yield of EVs endogenously from MSCs for clinical use [26]. In recent studies, in order to obtain high yields of EVs, human cells were treated with cytochalasin B and stimulated to detach membrane vesicles by shaking. These cytochalasin B-induced membrane vesicles (CIMVs) had functionally active surface receptors and contained cytoplasmic proteins representative of the parental cells [36]. Application of cytochalasin B to human cells significantly increases the yield of membrane vesicles. The protein concentration of EVs produced endogenously by 10^8^ MSC is 402.48 ± 178.12 μg, whereas the protein concentration of CIMVs obtained from 10^6^ MSC is about 1 mg [37]. This method of CIMV isolation can overcome the problem of limited vesicle yield and simplify the vesicle production procedure, since it involves only a few low speed centrifugation steps.

Most commonly, the size of natural EVs isolated from MSCs ranges between 60 nm and 150 nm, corresponding to exosome-like vesicles, with only a small number of EVs ranging in size from 200 nm to 400 nm, corresponding to larger EVs [38,39]. The majority of isolated CIMVs (about 70%) had a size of less than 220 nm, which again corresponds to the size of the smaller exosome-like vesicles produced endogenously. Endogenous EVs isolated from various cell types, including MSCs, contain damaged and functional components of the mitochondrial network [40,41,42]. We showed that almost a quarter (about 23%) of all CIMVs also contained a mitochondrial component. In different studies of endogenous EVs, the number of vesicles containing mitochondria varies, for example myeloid-derived regulatory cells (MDRC)-derived EVs contained the mitochondrial component in 33 ± 19.7% of cases [43], 25% of MSCs-isolated EVs carried mitochondria [44], while proteomic analysis of human vesicular proteins showed a mitochondrial content of 10% [45]. In general, our CIMVs contained mitochondria within the range similar to endogenous EVs. EVs also contain fragments of RNA, as well as various types of DNA, including nuclear DNA [46]. Available data indicate that up to 90% of endogenous EVs contain fragments of nuclear DNA [47]. The obtained CIMVs contained a nuclear component in about 7% of cases, which is due to the fact that when using cytochalasin B to isolate EVs, the nuclei are not destroyed and are precipitated during the second centrifugation step [48].

We have also analyzed the expression level of common MSC surface markers on the surface of hADSCs and CIMVs isolated from them. Normally, human MSCs isolated from adipose tissue are strongly positive for CD 29, CD44, CD73, CD90 and CD166 markers and negative for hematopoietic (CD45, CD133, CD117) markers [49]. The isolated CIMVs also expressed a limited number of hADSC surface receptors. The same results have been shown in other studies of MSC-isolated EVs, which also expressed a small number of parental cell receptors [38,39]. At the same time, genetic modification had no effect on the number of CD markers, both on the surface of hADSCs and on the surface of CIMVs.

The therapeutic effect of IL2 is mediated by its ability to regulate growth as well as differentiation of T-cells, B cells, natural killer cells, and many other cell types [50]. In our previous work, we showed that the conditioned medium from hADSCs-IL2 can activate CD8^+^ T-cells, as well as CD56^+^ NK cells [51]. Therefore, we decided to evaluate the effect of hADSCs-IL2, as well as CIMVs-IL2 isolated from them, on various populations of immune cells isolated from the peripheral blood of healthy adults.

CD8^+^ T-cells are an effector group of T-cells (T-killers), which play an important role in the anti-tumor immune response, as they can kill tumor cells by granule exocytosis and Fas ligand (FasL) (CD95)-mediated apoptosis [52]. CD8^+^ T-cells, that simultaneously express CD38 and HLA-DR activation markers are considered as activated T-cells that exhibit high effector functions, such as proliferation, cytotoxicity, and cytokine production [53]. We found that cultivation of CD8^+^ T-cells with both hASDCs-IL2 and with CIMVs-IL2 led to a significant increase in the number of activated CD38^+^HLA-DR^+^ T-killers. However, cultivation of PBMCs with CIMVs-IL2 led to a more significant (by two times) increase in the number of activated T-killers compared to the hADSCs (by 1.3 times). It has also been shown that the administration of IL15, which has many overlapping functions with IL2, leads to an increase in the expression of CD38 and HLA-DR activation markers on the surface of CD8^+^ T-cells [54]. MSCs have been shown to be able to downregulate other markers of CD8^+^ T-cell activation (such as CD44 and CD95) [55]. However, we, on the contrary, found a slight increase in the number of CD38^+^HLA-DR^+^ T-killers, but it was not as strong as in the hADSCs-IL2 sample.

NK cells play a pivotal role in the anti-tumor immunity, which can induce apoptosis of cells that have abnormal or altered major histocompatibility complex (MHC) I expression [56]. Typically, NK cells express CD56, in the absence of CD3 (T-cell receptor). We did not observe an IL2-associated increase in the number of CD56^+^CD3^−^NK cells in the PBMC population after the cultivation with hADSCs-IL2, although cultivation with the CM from hADSCs-IL2 resulted in an increase in the number of NK cells [51]. Moreover, cultivation of PBMCs with CIMVs resulted in a significant decrease in the number of NK cells. This suppressive effect is likely mediated by the negative effect of CIMVs on the proliferation of NK cells, since it has previously been shown that EVs isolated from MSCs suppress the proliferative activity of PBMCs [38].

IL2-based therapy has several significant disadvantages, particularly as it can induce anti-tumor immune response through Treg expansion [17]. Stimulation with IL2 leads to increased expression of CD25 [57] on the surface of CD3^+^ T-cells and the formation of the Tregs phenotype, including during low-dose therapy [58]. Our data show that cultivation of PBMCs with hADSCs led to an increase in the number of CD4^+^CD25^+^CD127^low^ Tregs in all the hADSC cultivated samples by about 25%. This increase is most likely associated not so much with the IL2 overexpression, as with the ability of MSCs to increase in the counts and frequency of CD4^+^CD25^high^Foxp3^+^CD127^low^ Treg cells [59,60]. However, the co-culture of PBMCs with both native CIMVs, CIMVs-BFP and CIMVs-IL2 did not result in a significant increase in Tregs compared to native PBMCs.

Other populations of Th cells, such as Th1, Th2 and Th17, also play an important role in the anti-tumor immune response. Th1 can enhance the priming and expansion of CD8^+^ T-cells or inhibit angiogenesis in IFN-γ-dependent manner [61,62]. The effect of Th2 and Th17 on tumor growth is controversial [63,64]. However, we decided to analyze the change in the distribution of Th cell populations after interaction with hADSCs and CIMVs. As previously described, MSCs can suppress Th17 differentiation from both naïve- and memory-phenotype precursors [65]. The same properties have been described for EVs isolated from mesenchymal stem cells [66,67]. The CIMVs isolated by us reduced the number of Th17 cells relative to the rest of the T helper population. However, the incubation of PBMCs with genetically modified hADSCs, on the contrary, led to an increase in the number of Th17 cells. We believe this is due to the fact that genetic modification with both BFP and IL2 leads to an increase in the secretion of the TGF-β1 protein, as has been shown in our previous study. TGF-β1 plays an important role in the maturation of naive T-cells in Th17 cells [68], and probably the constant increased secretion of this protein by hADSCs-BFP and hADSCs-IL2 led to an increase in the number of Th17 cells [51]. In the case of CIMVs, the amount of TGF-β1 carried in the vesicles had no significant effect on Th17 cells, and we detected an inhibitory effect of MSC-derived vesicles. Small changes (both an increase and a decrease) in the Th1 and Th2 number were observed in various samples treated with both native and genetically modified hADSCs. However, we were unable to identify a strong trend in changes of the number of Th1 or Th2 cells after cultivation with hADSCs or CIMVs. 

Therefore, we observed that the hADSCs-IL2 and CIMVs-IL2 can stimulate the activation of CD8^+^ T-killers. However, different levels of immunomodulatory effects may be due to the natural ability of MSCs to suppress T-cell proliferation [32,69]. Therefore, we analyzed the effect of hADSCs and CIMVs on T-cell proliferation. For this, the proliferation of CD4^+^ or CD8^+^ T-cell was stimulated with PHA, a mitogen that stimulates T-cell division [70]. At 72 h, the proliferation of both PHA-induced CD8^+^ T-cell and CD4^+^ T-cell was significantly suppressed when co-cultured with any of the hADSC lines compared to the proliferation of PHA-only induced T-cells cultured alone. These data are consistent with previous studies indicating an immunosuppressive effect of MSCs. However, the cultivation of PHA-stimulated CD8^+^ T-cell and CD4^+^ T-cell with CIMVs failed to affect the proliferation of immune cells. Similar results were obtained in other studies, where endogenous EVs isolated from MSCs also did not have an inhibitory effect on T-cell proliferation [38]. However, we did not observe any IL2-mediated stimulation of T-cell proliferation, so we decided to stimulate T-cells in a different way. It is known that the activation of naïve CD8^+^ T-cells begins with the binding of CD3 on the T-cell surface with the MHCI-protein complex on the surface of antigen presenting cell (APC). In addition, CD28 replaced on the surface of CD8^+^ T-cells recognizes co-stimulatory B7 proteins (CD80 and CD86) of APC. After the antigen presentation, CD8^+^ T-cells begin to secrete IL2 and also express its receptor (CD25), thus stimulating proliferation by themselves [71]. Therefore, we decided to mimic antigen presentation by adding CD3 and CD28 antibodies and analyse the effect of hADSCs and CIMVs on T-cell proliferation. The division of PBMCs after the addition of CD3/CD28 was weak, but the proliferation of both CD8^+^ T-cells and CD4^+^ T-cells was significantly increased after cultivation with hADSCs-IL2. At the same time, the proliferation of CD8^+^ T-cells after cultivation with CIMVs-IL2 did not significantly increase, and a slight increase was observed in the population of CD4^+^ T-cells. It is likely that more significant results obtained after cultivation with hADSCs-IL2 are explained by the large amount of IL2, which is constantly secreted from cells for 72 h.

The obtained data on the effect of hADSCs-IL2 and CIMVs-IL2 on the activation and proliferation of PBMCs suggest that CIMVs-IL2 can activate cytotoxic T-killers much more efficiently, and at the same time do not suppress the proliferation of already dividing T-cells, in contrast to hADSCs-IL2. However, the ability of CIMVs-IL2 to stimulate cell proliferation after antigen presentation is much less noticeable. We assume that this is due to the higher concentration of IL2, which is constantly secreted by hADSCs-IL2 for 72 h of co-cultivation. The IL2-associated effect is not found in the PHA experiment, since the immune cell proliferation is already stimulated without antigen presentation and the cells do not produce more receptors for IL2, which can stimulate their proliferation, as is the case with the experiment with CD3+CD28 antibodies. It is likely that hADSCs-IL2 can more effectively support the proliferation of T-cells after antigen presentation. However, secretion of other immunosuppressive factors by hADSCs makes CIMVs-IL2 a much more defined therapeutic tool compared to hADSCs-IL2.

In order to evaluate our hypothesis regarding the therapeutic efficacy of CIMVs, we analyzed the cytotoxic effect of hADSC- and CIMV-activated PBMCs on triple negative breast cancer (TNBC) cells. Breast cancer is the most commonly diagnosed cancer and the second leading cause of cancer death among women worldwide [72]. One of the challenges in the treatment of breast cancer is tumor heterogeneity, which determines the treatment options [73]. TNBC is the most aggressive and difficult to treat breast cancer subtype, since hormone receptors (HR) are absent and the human epidermal growth factor receptor 2 (HER2) protein is not overexpressed on the surface of TNBC cells. The lack of HR and HER2 expression prevents the use of targeted therapies that are effectively applied to other breast cancer subtypes. This leaves chemotherapy as the only approved option for systemic TNBC treatment. However, the frequency of relapses and metastases of triple negative breast cancer is very high, and the mean overall survival rate for patients with metastatic TNBC is about 9–12 months, when treated with conventional cytotoxic agents [74]. IL2 has been used to treat TNBC patients in combination with low-dose (LD) immune checkpoint blockade [75], as well as Cetuximab [76] in order to increase the effectiveness of the immune response, with varying degrees of success. For example, administration of high-dose interleukin-2 in combination with lymphocyte-activated killer cells was ineffective in patients with breast cancer [77]. At the same time, the effect of IL2-based immunotherapy alone has not been evaluated. Twenty-four hours after the addition of PBMCs activated with CIMVs-IL2, the proliferation of MDA-MB-231 cells was decreased by four-fold. This decrease in proliferation is likely due to a proportion of the tumor cells undergoing apoptosis as a result of the actions of CIMV-IL2-activated PBMCs. This is confirmed by the MDA-MB-231 cell viability test, after 24 h of cultivation with PBMCs activated with CIMVs-IL2. In the experiment with hADSCs-IL2, a decrease in the proliferation and apoptosis of MDA-MB-231 cells after 24 h of cultivation were also observed, but they were not as significant as in the experiment with CIMVs-IL2. Since activated T-killers, the number of which was increased after cultivation with CIMVs-IL2, are able to destroy cancer cells [78], including breast cancer cells [79], we believe that a higher number of activated T-killers after incubation with CIMVs-IL2 had a stronger cytotoxic effect on MDA-MB-231 cells and could cause the death of a higher number of tumor cells. However, surviving MDA-MB-231 cells in the sample of PBMCs after both CIMV-IL2 and hADSCs-IL2 continued to proliferate, however their proliferative activity remained reduced after 72 h of cultivation. The increased cytotoxicity of CIMVs-IL2-activated PBMCs was also confirmed after cultivation with MDA-MB-436 where the number of healthy cancer cells was decreased by 17% in the sample of cancer cells cultured with PBMCs activated with hADSCs-IL2, and by 28% in the sample of cancer cells cultured with PBMCs activated with CIMVs-IL2.

Since CIMVs-IL2 were able to exert an effect on CD8^+^ T-killers, we decided to analyze their impact on murine PBMCs in vivo. The choice of mice for the in vivo experiment was due to the fact that human and mouse IL2 show 57% sequence homology, and human IL2 is able to bind with murine αβγ-IL2 receptors [80,81]. Unfortunately, we did not find that the administration of CIMVs-IL2 affected the number of CD45^+^CD3^+^CD4^−^CD8^+^ T-cells or the CD45^+^CD3^+^CD8^−^CD4^+^ T-cells murine PBMC population. We noted a trend towards a decrease in the number of both CD4^+^ T-cells as well as CD8^+^ T-cells after the administration of CIMVs, and a trend towards an increase in the number of CD4^+^ T-cells and CD8^+^ T-cells in the sample of murine PBMCs after the injection of CIMVs-IL2. However, no statically significant difference was found. This could probably be due to the fact that human IL2 shows 10-fold lower affinity for the murine receptor expressed on T lymphocytes than murine IL2 [82], and as such the number of injected recombinant human IL2 (rhIL2) or CIMVs was unable to induce a significant immune response. In order to better evaluate the effect of CIMVs-IL2 on the mouse immune response, the levels of cytokines/chemokines in the serum of mice 72 h after the injection of CIMVs were assessed. The secretion of the pro-inflammatory cytokines IL3, IL13 and the chemokine MCP-1 were elevated following administration of native CIMVs and CIMVs-BFP compared to the PBS control group. The level of these proteins remained unchanged after the injection of CIMVs-IL2, showing no significant difference from the control group that received PBS only. IL-3 is mostly produced by T-cells upon activation and contributes to ongoing inflammation by modulating monocyte, macrophage and eosinophil proliferation and activation [83]. IL13, which is synthesized by Th2 cells, NKT-cells and other immune cells, also plays a role in the development and maintenance of inflammation, and is associated with autoimmune diseases in both mice and humans [84,85]. MCP-1 functions to recruit monocytes and NK cells into foci of active inflammation [86]. Increased secretion of pro-inflammatory cytokines in the mice injected with native CIMVs and CIMVs-IL2 may indicate an inflammatory response of the immune system. This is further supported by the observed trend towards an increase in the levels of pro-inflammatory cytokines IFN-γ, IL2, IL5, IL12p40 and MIP-1α in the mouse serum after the administration of native CIMVs or CIMVs-BFP, but not CIMVs-IL2. However, the level of G-CSF, which stimulates the proliferation of monocytes in response against viral and bacterial infections, was reduced in mice treated with native CIMVs and CIMVs-BFP [87]. This confirms that the inflammatory response was induced by the administration of native CIMVs and CIMVs-BFP, and not via an infection. Interestingly, that same effect was not observed after the injection of CIMVs-IL2. It is also worth noting that the level of eotaxin, which is a chemoattractant for eosinophils in autoimmune inflammation [88], was significantly reduced after the administration of CIMVs-IL2 compared to injection with the other CIMV samples, confirming that CIMVs-IL2 do not induce an autoimmune response. The mechanisms of this anti-inflammatory effect of CIMVs-IL2 requires more detailed examination, alongside the anti-tumor properties of CIMVs-IL2.

## 5. Conclusions

CIMVs-IL2 were more able to activate cytotoxic CD8^+^ T-killers than hADCs-IL2, and did not suppress the proliferation of immune cells, unlike hADSCs. CIMVs-IL2-activated T-killers were found to be able to effectively kill human triple negative cancer MDA-MB-231 and MDA-MB-436 cells. The effectiveness of the use of CIMVs-IL2 remains to be proven in humanised cancer mouse models with reconstituted human immune system, as the next step to realising the clinical potential of therapeutic CIMVs.

## Figures and Tables

**Figure 1 biology-10-00141-f001:**
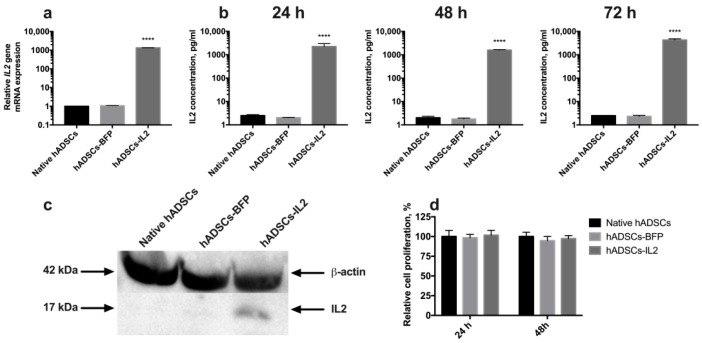
Detection of interleukin 2 (IL2) expression in human adipose tissue-derived mesenchymal stem cells (hADSCs). (**a**) Relative expression of the IL2 gene mRNA in hADSCs-IL2 was 1000-fold increased. The data were normalized using an 18S ribosomal RNA housekeeping gene. Bars represent the mean of two biological replicates with their corresponding standard deviation (SD) (*n* = 6). (**b**) The secretion of IL2 protein was increased 2000-fold in hASDCs-IL2 compared to native hADSCs and hADSCs-blue fluorescent protein (BFP) after 24 h of cultivation. This significant increase in IL2 secretion by hADSCs-IL2 was maintained when analyzed in media conditioned for 48 and 72 h. (**c**) Expression of IL2 protein was confirmed using Western blot analysis with β-actin serving as an internal control. (**d**) Relative proliferation of hADSCs at 24- and 48-h post seeding in vitro has no statistically significant difference between cell cultures. Each value is presented as the % in relation to the control (native hADSCs) group. Bars represent the mean ± SD (*n* = 6) of two biological replicates. **** *p* < 0.0001.

**Figure 2 biology-10-00141-f002:**
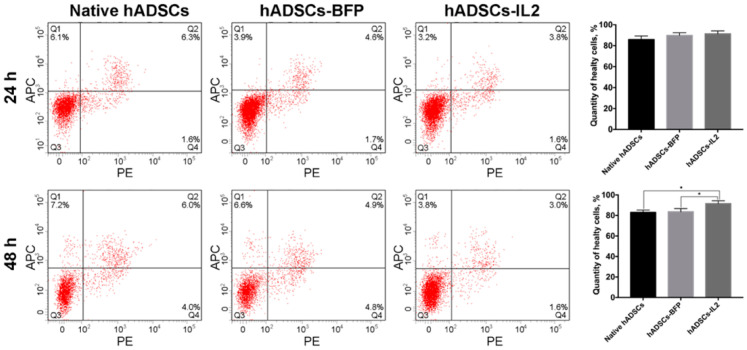
Viability analysis of native and genetically modified hADSCs. Cells were cultured for 24 and 48 h and stained with Annexin/PI to detect apoptosis and necrosis. Data were obtained using flow cytometry. Q1: number of apoptotic cells. Q2: number of late-apoptotic cells. Q3: number of healthy cells. Q4: number of necrotic cells. Each box represents the mean ± SD (*n* = 6) of two biological replicates of the number of healthy native hADSCs, hADSCs-BFP or hADSCs-IL2. * *p* < 0.05.

**Figure 3 biology-10-00141-f003:**
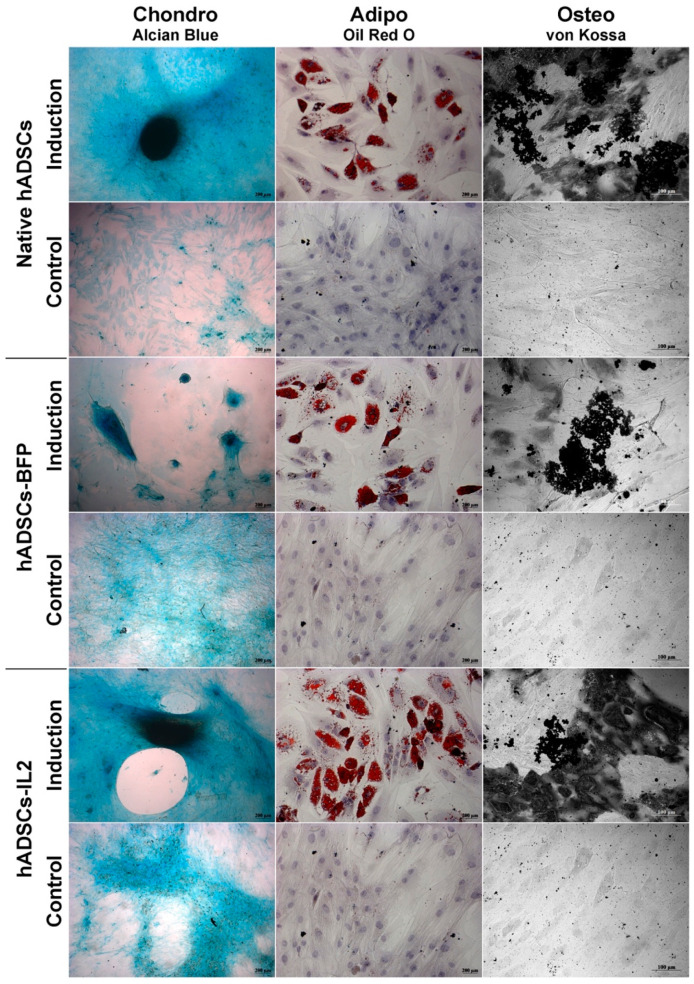
Chondrogenic, adipogenic and osteogenic differentiation of hADSCs. Prior to image analysis by phase contrast microscopy, cells were differentiated as follows. Chondrogenic differentiation was determined by staining with Alcian blue on day 21 after seeding. For adipogenic differentiation, native and IL2-genetically modified cells were cultured in the reprogramming medium for 14 days. At day 14, cells were fixed and stained with Oil Red O. For osteogenic differentiation, cells were cultured in the reprogramming medium for 28 days. At day 28, cells were fixed and analyzed by von Kossa staining (chondrogenic and adipogenic differentiation: scale bar, 200 µm osteogenic differentiation: scale bar, 100 µm).

**Figure 4 biology-10-00141-f004:**
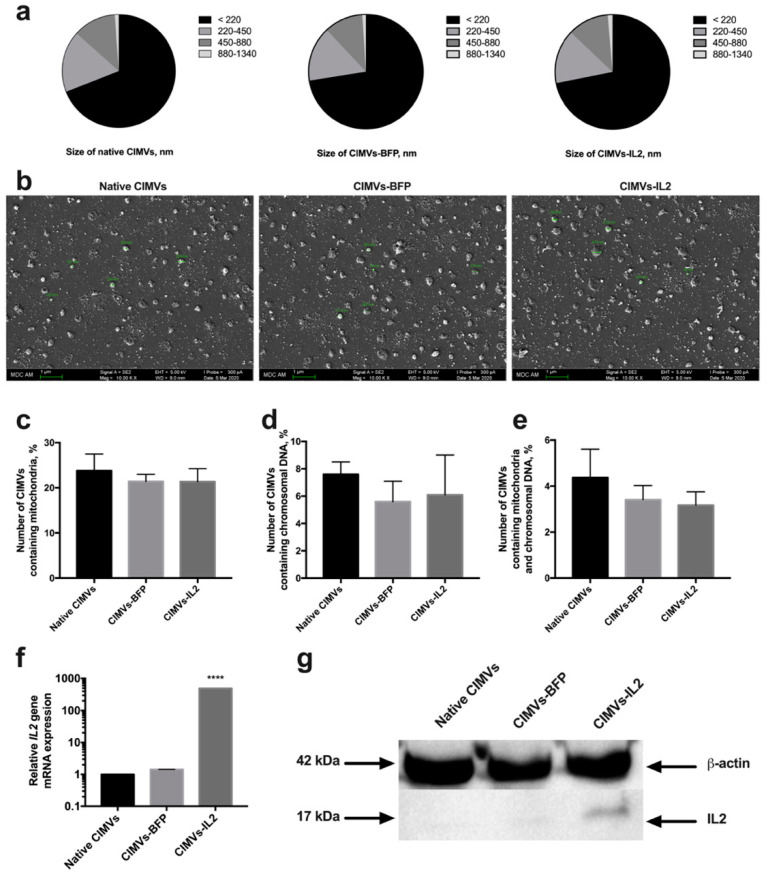
The size of cytochalasin B-induced membrane vesicles (CIMVs) was analyzed using flow cytometry (**a**) with calibrating beads as well as scanning electron microscopy (**b**) finding the majority of the CIMVs were less then 220 nm in diameter. The scale bar is 1 µm. Analysis of the presence of the mitochondrial component showed the presence of mitochondria stained with MitoTracker Green in 23% of the isolated CIMVs (**c**). The presence of a nuclear component stained with the Hoechst 33,342 vital fluorescent dye was detected in 6% of the isolated CIMVs (**d**). The number of CIMVs containing both nuclear and mitochondrial components was about 4% (**e**). The presence of the IL2 gene mRNA (**f**) and IL2 protein (**g**) in isolated CIMVs was confirmed using qPCR and Western blot, accordingly. Each box represents the mean ± SD (*n* = 6) of two biological replicates. **** *p* < 0.0001.

**Figure 5 biology-10-00141-f005:**
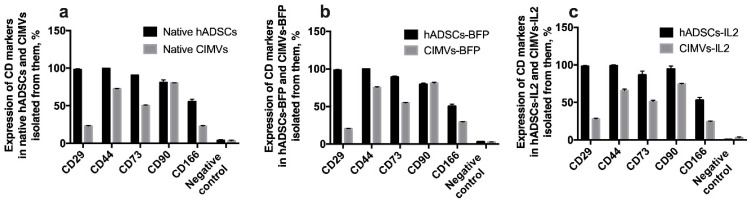
Immunophenotype of native hADSCs (**a**), hADSCs-BFP (**b**) hADSCs-IL2 (**c**) and CIMVs isolated from each cell line. Cells and CIMVs were analyzed using staining with antibodies typical for CD markers of MSCs detected by flow cytometry. The expression of typical surface MSC markers was significantly reduced in both native CIMVs (**a**), and genetically modified CIMVs-BFP (**b**) and CIMVs-IL2 (**c**). Each box represents the mean ± SD (*n* = 6) of two biological replicates. Percentages on graphs represent cells positive for the markers from the parent population of all hADSCs or CIMVs.

**Figure 6 biology-10-00141-f006:**
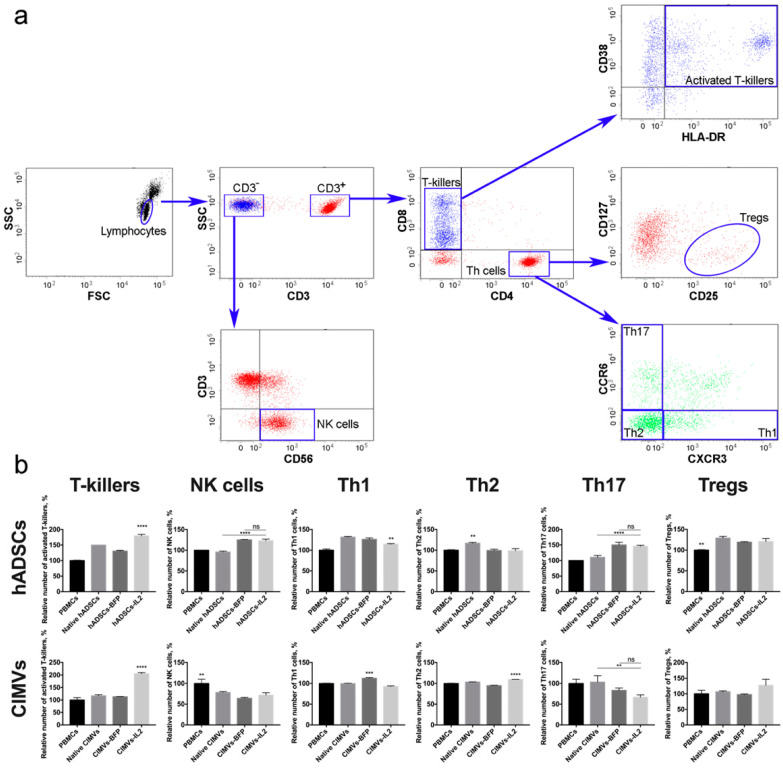
(**a**) Gating strategy for flow cytometry of immune cells to analyse the following populations: CD3^+^ CD8^+^ CD4^−^ CD38^+^ HLA-DR^+^ T-killers, CD3^−^CD56^+^ NK cells, CD3^+^CD4^+^CD8^−^CCR6^−^CXCR3^+^ Th1 cells, CD3^+^CD4^+^CD8^−^CCR6^−^CXCR3^−^ Th2 cells, CD3^+^CD4^+^CD8^−^CCR6^+^CXCR3^−^ Th17 cells, CD3^+^CD4^+^CD8^−^CD25^+^CD127^low^ Tregs. (**b**) The cultivation of CD8^+^ T-cells with hASDCs-IL2 and CIMVs-IL2 led to a significant increase in the number of activated CD38^+^HLA-DR^+^ T-killers. However, the cultivation of peripheral blood mononuclear cells (PBMCs) with CIMVs-IL2 led to a more significant (2-fold) increase in the number of activated T-killers compared to hADSCs-IL2 cells (by 1.3 times). The number of NK cells did not increase after culturing with hADSCs, and decreased after incubation with CIMVs. Small changes (both an increase and a decrease) in the number of Th cells were observed in various samples, both native and genetically modified. At the same time, CIMVs-IL2 did not increase the number of regulatory T-cells (Tregs), unlike hADSCs-IL2. Each box represents the mean ± SD (*n* = 4) of four biological replicates, the data were calculated as relative to control PBMCs. ***p* < 0.01, ****p* < 0.001, *****p* < 0.0001.

**Figure 7 biology-10-00141-f007:**
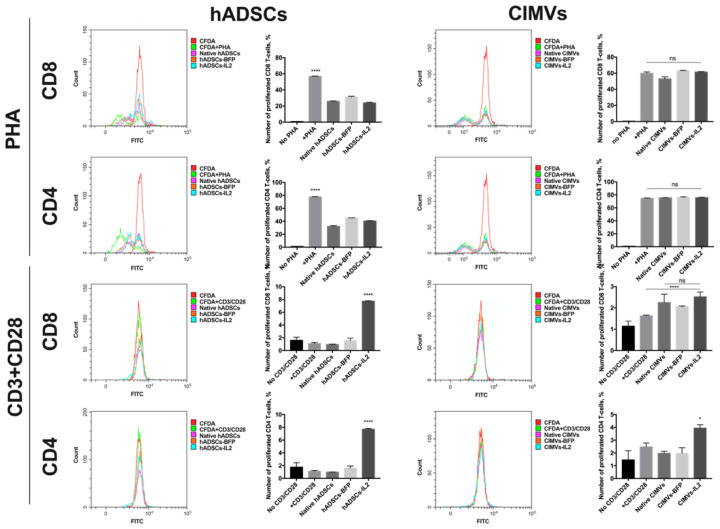
In order to evaluate the effect of hADSCs and CIMVs on the proliferative activity of T-cells, the proliferation of PBMCs isolated from peripheral blood was stimulated using PHA or a CD3 + CD28 antibody complex to mimic antigen presentation. Analysis of PHA-stimulated T-cell proliferation after co-culture with hADSCs showed that co-cultivation with hADSCs had a significant suppressive effect on the proliferation of CD4^+^ and CD8^+^ T-cells. Whilst the proliferation of PHA-stimulated CD4^+^ and CD8^+^ T-cells remained unchanged after co-culture with CIMVs. After stimulation with CD3 + CD28 antibodies, an increase in the number of CD4^+^ and CD8^+^ T-cells was detected in the PBMCs after cultivation with hADSCs-IL2, while after cultivation with CIMVs-IL2, the number of CD4^+^ T-cells slightly increased, and the number of CD8^+^ T-cells remained unchanged. Each histogram represents the fluorescence of PBMCs after cultivation with hADSCs or CIMVs, where the weaker the fluorescence intensity in comparison with control culture (designated as CFDA), the higher the proliferative activity. Each graph represents the number of proliferating cells which was calculated in reference to negative control. The negative control, in which PBMCs remained unstimulated (no PHA or CD3+CD28 was added), was used to define the threshold of CFDA signal for non-proliferating T-cells. The number of cells with lower CFDA per cell (as compared to the negative control) was accepted as the number of proliferating cells. Each box represents the mean ± SD (*n* = 4) of four biological replicates. * *p* < 0.05, **** *p* < 0.0001.

**Figure 8 biology-10-00141-f008:**
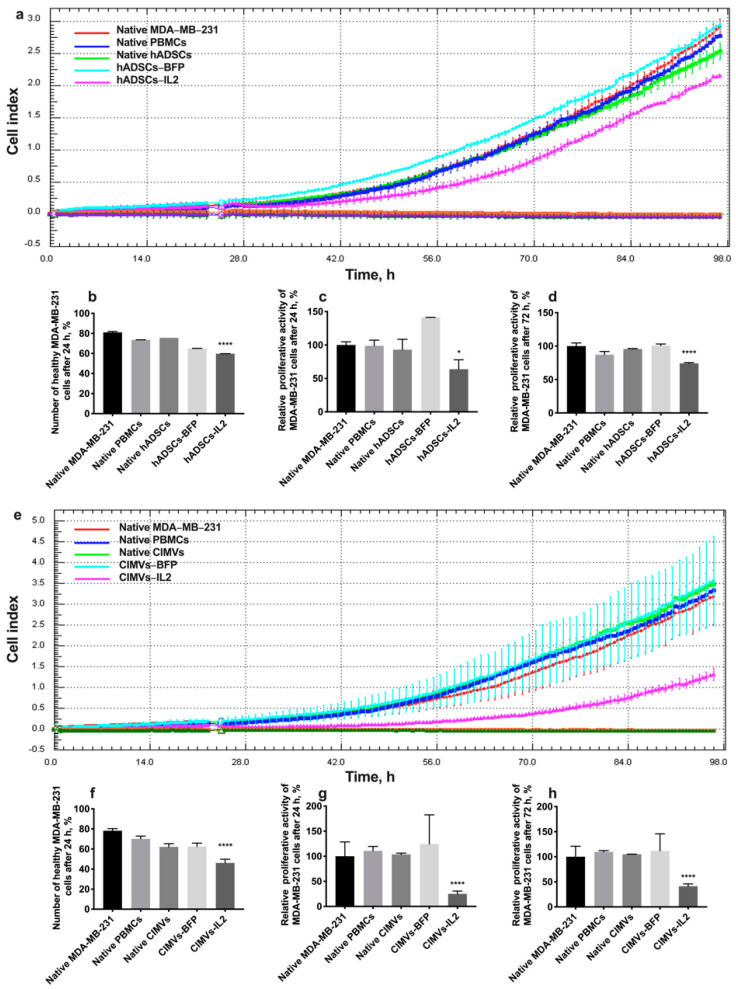
In order to assess the cytotoxic activity of hADSC- or CIMV-activated PBMCs against MDA-MB-231 triple negative breast cancer cells, proliferative activity and the number of apoptotic tumor cells were determined. After the addition of PBMCs activated with hADSCs-IL2, the proliferation of MDA-MB-231 cells was decreased by 36% at 24 h (**a**,**b**) and by 26% at 72 h (**a**,**d**). The number of nonapoptotic and non-necrotic MDA-MB-231 cells was also decreased after cultivation with hADSCs-IL2-activated PBMCs (**c**). After the addition of PBMCs activated with CIMVs-IL2, the proliferation of MDA-MB-231 cells was significantly decreased. Twenty-four hours after the addition of CIMV-IL2-activated PBMCs the proliferation of MDA-MB-231 was decreased 4-fold (**e**,**f**). The number of healthy MDA-MB-231 cells after 24 h of cultivation with PBMCs activated by CIMVs-IL2 was reduced by 35% (**g**). The proliferative activity of MDA-MB-231 cells after cultivation with CIMVs-IL2-activated PBMCs remained reduced by 59% after 72 h of cultivation (**e**,**h**). Each proliferation graph represents the changes in cells index (a unitless parament which indicates the impedance measured through gold electrodes on which the cells are seeded) over time. Each box represents the mean ± SD (*n* = 4) of four biological replicates. Proliferative activity is calculated as % relatively to native MDA-MB-231 cells. The cell viability graphs (**b**,**f**) represent the absolute value of non-apoptotic/non-necrotic (healthy) cells found in the cell population. * *p* < 0.05, **** *p* < 0.0001.

**Figure 9 biology-10-00141-f009:**
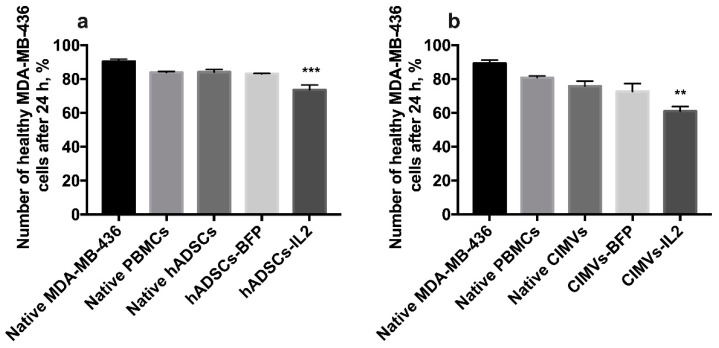
The cytotoxicity of activated PBMCs was also assessed in MDA-MB-436 cells. The cultivation of MDA-MB-436 cells with hADSCs-IL2-activated PBMCs resulted in the decrease of healthy (non-apoptotic and non-necrotic) cancer cells by 17% (**a**). The number of healthy MDA-MB-436 cells after 24 h of cultivation with PBMCs activated by CIMVs-IL2 was also reduced by 28% (**b**). Each box represents the mean ± SD (*n* = 4) of four biological replicates. ** *p* < 0.01, *** *p* < 0.001.

**Figure 10 biology-10-00141-f010:**
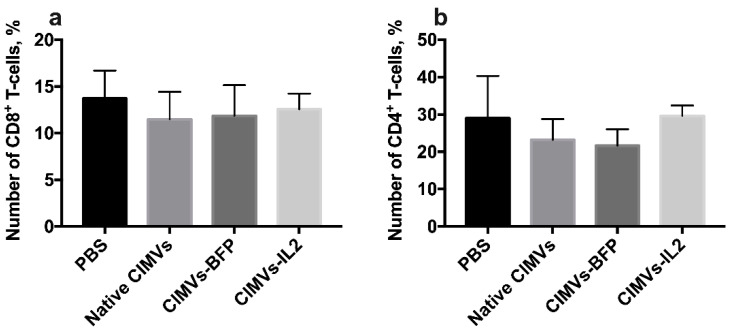
To analyze the effect of CIMVs-IL2 in in vivo, CD-1 immunocompetent mice were intravenously injected with 50 µg (in 200 µL of PBS) of native CIMVs, CIMVs-BFP or CIMVs-IL2. After 72 h there was no statistically significant difference in the number of murine CD45^+^CD3^+^CD4^−^CD8^+^ T-cells (**a**) as well as CD45^+^CD3^+^CD8^−^CD4^+^ T-cells (**b**) after cultivation with CIMVs-IL2. Each box represents the mean ± SD (*n* = 5) of five biological replicates, which are calculated as relative to control PBS group.

**Figure 11 biology-10-00141-f011:**
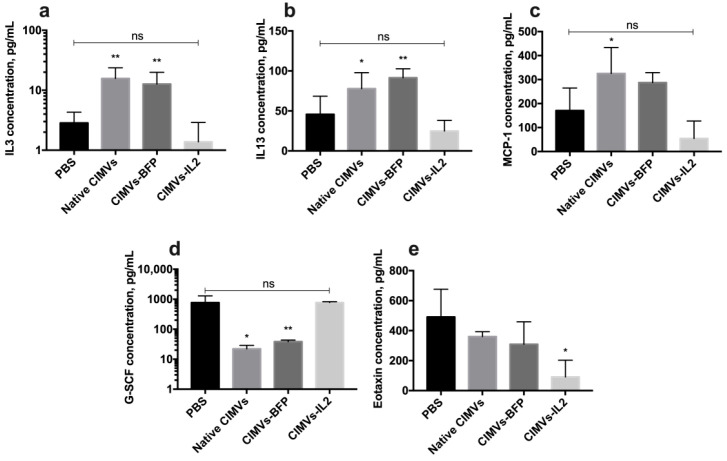
Changes in the cytokine profile of mice after injection of native CIMVs, CIMVs-BFP or CIMVs-IL2 using the Bio-Plex Pro Mouse Cytokine 23-plex panel. After 72 h, the levels of pro-inflammatory cytokines IL3 (**a**), IL13 (**b**) and monocyte chemoattractant protein 1 (MCP-1) (**c**) were significantly increased in the mice injected with native CIMVs and CIMVs-BFP compared to the PBS group. At the same time, in mice injected with CIMVs-IL2, those cytokine levels remained similar to the PBS control group. The level of granulocyte colony-stimulating factor (G-CSF) (**d**), which is involved in the immune response to viral and bacterial infections, was reduced in mice treated with native CIMVs and CIMVs-BFP compared to CIMVs-IL2 or PBS only. The secretion of eotaxin (**e**), which is a chemo attractant for eosinophils in autoimmune inflammation, was re-duced after the administration of CIMVs-IL2 compared to injection with the other CIMVs or PBS. Each box represents the mean ± SD (n = 4) of four biological replicates. *p* < 0.05, * *p* < 0.01, ns—not significant.

**Table 1 biology-10-00141-t001:** Primer and probe sequences of related genes for quantitative polymerase chain reaction (qPCR).

Target Gene	Forward Primer (5′–3′)	Reverse Primer (5′–3′)	TaqMan Probe (5′–3′)
18S rRNA	GCCGCTAGAGGTGAAATTCTTG	CATTCTTGGCAAATGCTTTCG	[HEX] ACCGGCGCAAGACGGACCAG [BH2]
IL2	CACCAGGATGCTCACATTTAAG	GTCCCTGGGTCTTAAGTGAAAG	[FAM] CCCAAGAAGGCCACAGAACTGAAACA [BH1]

**Table 2 biology-10-00141-t002:** The levels of cytokines/chemokines in the serum of mice 72 h after injection of 50 µg (in 200 µL of PBS) of native CIMVs, CIMVs-BFP or CIMVs-IL2.

Protein	Protein Concentration in Mouse Serum, pg/mL			
	PBS (*n* = 5)	Native CIMVs (*n* = 5)	CIMVs-BFP (*n* = 5)	CIMVs-IL2 (*n* = 5)
GM-CSF	30.4 ± 6.9	69.5 ± 16.7	57.8 ± 31.8	31.8 ± 17.4
IFN-γ	27.1 ± 9.1	51.2 ± 19.4	43.9 ± 27.0	16.0 ± 1.0
IL10	40.1 ± 52.7	21.5 ± 6.9	24.1 ± 8.8	17.4 ± 13.4
IL2	10.6 ± 2.5	21.9 ± 12.8	20.2 ± 9.7	6.4 ± 0.5
IL4	13.5 ± 3.9	0.1 ± 0.1	14.0 ± 12.4	9.8 ± 2.7
IL5	0.2 ± 0.3	17.2 ± 8.9	5.9 ± 9.1	0.2 ± 0.3
IL6	7.5 ± 3.4	2.8 ± 1.1	8.9 ± 4.9	6.3 ± 4.0
IL12p40	37.2 ± 6.9	142.1 ± 59.5	87.9 ± 70.2	11.1 ± 2.2
IL12p70	182.4 ± 49.1	39.7 ± 11.1	112.1 ± 92.6	91.4 ± 71.5
KC	75.3 ± 31.0	106.7 ± 20.8	120.7 ± 14.0	48.9 ± 18.9
MIP-1α	5.8 ± 1.9	9.2 ± 3.4	9.2 ± 2.0	5.5 ± 0.6
MIP-1δ	12.4 ± 8.0	32.0 ± 15.6	28.8 ± 23.7	8.2 ± 1.9
RANTES	55.0 ± 19.7	42.5 ± 10.6	28.1 ± 19.3	38.2 ± 26.5
TNF-α	61.9 ± 14.8	88.3 ± 48.7	99.0 ± 51.4	68.0 ± 41.9

## Data Availability

Authors can confirm that all relevant data are included in the arti-cle and/or its supplementary information files.

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
