# Peer review of "Cytochalasin B-Induced Membrane Vesicles from Human Mesenchymal Stem Cells Overexpressing IL2 Are Able to Stimulate CD8+ T-Killers to Kill Human Triple Negative Breast Cancer Cells"

_biology, 2021, doi:10.3390/biology10020141_

Round 1

Reviewer 1 Report

The manuscript entitled as “Cytochalasin B-induced Membrane Vesicles ……. Triple Negative Breast Cancer 4 Cells. This paper well formulated, designed and written, but potentially failed to provide a definite conclusion  

Major comments

  1. Stability of IL-2 is missing in the conditioned medium and IL-2 overexpressed hADSCs cells
  2. In Figure. 1a there is a huge difference in the IL-2 gene expression, but in protein result, it was very low. How can you explain?
  3. In Figure. 2 a and b, how we can see same percentage of late apoptotic cells in the 24- and 48-hours viability assay?
  4. Change the figure legends order and details according to the figure. 3,
  5. In Figure. 4, the Hoechst 33342 stain data is missing
  6. Why this entire manuscript experiment in just one triple negative breast cancer cell MDA-MB-231?
  7. There is putatively lack of apoptosis and other data to prove 231 cell death
  8. There is a lack of significant data for the animal work, then how can you bring this into clinical treatment?
  9. This entire manuscript needs to check the statistical analysis because some results we can see a major difference, but the p value is a very significant one.
  10. Likewise, at least we need three independent biological repeats to conclude our hypotheses, but this manuscript most of the data represented as two biological replicates.
  11. Need to improve the western blot results
  12. In method section, they need to include protein loading concentration and loading control for the secreted protein in the medium
  13. Moreover, they must include other characteristic features of Membrane vesicle such as confocal or immune fluorescence images.
  14. They need to improve the graph for the cytotoxicity assay in the Figure. 8 a and e.
  15. They need to reduce the discussion section and most of them, nor relevant to the current findings.

This manuscript is potential lack of addition data to support the hypothesis and there is no further outcome for the clinical treatment aspects.

Author Response

Thank you for your valuable comments. We have corrected them point by point within the manuscript accordingly (your comments are in bold text and our responses are in ordinary type):

1) Stability of IL-2 is missing in the conditioned medium and IL-2 overexpressed hADSCs cells

Since we investigated the effect of IL2 on the activation of immune cells, which occurs over a short period of time, up to 72 hours, we have studied IL2 stability for short periods of time, and it was shown the protein was present in the conditioned medium after 72 hours of cultivation.

2) In Figure. 1a there is a huge difference in the IL-2 gene expression, but in protein result, it was very low. How can you explain?

During western blot, the protein was detected only in the lane of hADSCs-IL2 sample, no protein was detected in the samples of native hADSCs and hADSCs-BFP, which indicates a significantly higher expression of IL2 in genetically modified hADSCs-IL2. Analysis of IL2 secretion into conditioned media also confirmed a significantly higher IL2 level (4000 times higher after 72 hours of cultivation) in hADSCs-IL2 compared to native hADSCs and hADSCs-BFP. This is described in details in Page 11, Lines 466-473.

3) In Figure. 2 a and b, how we can see same percentage of late apoptotic cells in the 24- and 48-hours viability assay?

The number of apoptotic cells barely changes between 24 and 48 hours since the apoptosis detection kit we used allows assessing the number of cells actively undergoing apoptosis and not total cell death.

4) Change the figure legends order and details according to the figure. 3,

It was corrected.

5) In Figure. 4, the Hoechst 33342 stain data is missing

The information about Hoechst 33342 staining is shown in Figure 4 under the letter d. The presence of a nuclear component stained with the Hoechst 33342 vital fluorescent dye was detected in 6% of isolated CIMVs (Figure 4 legend, Page 15, Lines 546-547).

6) Why this entire manuscript experiment in just one triple negative breast cancer cell MDA-MB-231?

Breast cancer is the most commonly diagnosed cancer and the second leading cause of cancer death among women worldwide [72]. One of the challenges in the treatment of breast cancer is tumour heterogeneity, which determines the treatment options [73]. TNBC is the most aggressive and difficult to treat breast cancer subtype, since hormone receptors (HR) are absent and the human epidermal growth factor receptor 2 (HER2) protein is not overexpressed on the surface of TNBC cells. The lack of HR and HER2 expression prevents the use of targeted therapies that are effectively applied to other breast cancer subtypes. Leaving chemotherapy as the only approved option for systemic TNBC treatment. However, the frequency of relapses and metastases of triple-negative breast cancer is very high, and the mean overall survival rate for patients with metastatic TNBC is about 9–12 months, when treated with conventional cytotoxic agents [74]. For TNBC therapy, IL2 was used only in combination with other therapeutic agents, so we decided to analyze its therapeutic potential separately. We have added relevant information in the Discussion to make the choice of MDA-MB-231 cells more reasoned for the reader (Page 26, Line 969-980).

7) There is putatively lack of apoptosis and other data to prove 231 cell death

The number of apoptotic, late-apoptotic and necrotic MDA-MB-231 cells was analyzed 24 hours after cultivation with PBMCs using the APC Annexin V Apoptosis Detection Kit with PI. The data are presented in Figure 8 and on Page 19, Lines 714-717, 726-729 as the number of healthy cells (non-apoptotic and non-necrotic cells).

8) There is a lack of significant data for the animal work, then how can you bring this into clinical treatment?

This investigation was a proof of concept study to demonstrate an anti-tumor effect of the CIMVs-IL2. Since we had no observed immunomodulatory effect of human CIMVs with human IL2 in immunocompetent mice, in the future we are planning to create humanized cancer mouse models with reconstituted human immune system.

9) This entire manuscript needs to check the statistical analysis because some results we can see a major difference, but the p value is a very significant one.

All the obtained data were processed using one-way ANOVA followed by Tukey HSD post-hoc comparisons test. Significant probability values are denoted as * p < 0.05, ** p < 0.01 and *** p < 0.001, **** p < 0.0001.

10) Likewise, at least we need three independent biological repeats to conclude our hypotheses, but this manuscript most of the data represented as two biological replicates.

Yes, in the case of experiments with hADSCs, cells were isolated from two different donors and analyzed as two biological replicates. In every experiment there were three technical replicates which were summarized during data analysis. The information about the vesicles was obtained in the same way. However, in the experiments with PBMCs, PBMCs were isolated from four different donors and results were obtained in 1 replicate and summarized together during analysis. This is denoted as n = 4, four biological replicates. In the experiment with mice, the results of flow cytometry were also obtained in 1 replicate for one animal and summarized together (n = 5, five biological replicates).

11) Need to improve the western blot results

In order to support the Western blot results, we decided to determine the IL2 concentration per µg of total protein in hADSCs-IL2 and CIMVs-IL2 using ELISA. Methods (Page 8, Lines 314-321) and results (Page 11, Line 475; Page 15, Lines 554-555) were added to the manuscript.

12) In method section, they need to include protein loading concentration and loading control for the secreted protein in the medium

Loading concentration for Western blot was 40 µg of total protein for hADSCs and 50 µg of total protein for CIMVs. The appropriate information was added in the Materials and methods section (Page 6, Lines 223-224). Secreted protein was measured using cytokine multiplex analysis where IL2 standards ranged from 6438,84 pg/mL to 1,48 pg/mL, control concentration for IL2 was 83,59 pg/mL.

13) Moreover, they must include other characteristic features of Membrane vesicle such as confocal or immune fluorescence images.

We added confocal microscopy analysis of CIMVs in the supplementary materials (See FigureS2, Page 28, Lines 1053-1057).

14) They need to improve the graph for the cytotoxicity assay in the Figure. 8 a and e.

These graphs are built by the device and, unfortunately, the ability to edit them is limited.

15) They need to reduce the discussion section and most of them, nor relevant to the current findings.

We have tried to remove excessive information from the Discussion section.

16) This manuscript is potential lack of addition data to support the hypothesis and there is no further outcome for the clinical treatment aspects.

In order to get a more detailed view and draw conclusions regarding immunomodulation, we analyzed the cytokine profile of mice after the administration of CIMVs. We found an increase in the secretion of pro-inflammatory cytokines in the serum of mice treated with native CIMVs and CIMVs-BFP, but not CIMVs-IL2. The results are described and discussed in Page 21, Line 774 - Page 22, Line 811 and Page 26, Lines 1012-1039, respectively.

Reviewer 2 Report

  • Line 56-57: "The anticancer effect of high dose (HD) IL2 therapy was demonstrated by West et al. where 3 complete and 5 partial responses were reported in patients with metastatic renal cell carcinoma [2]." Out of how many?
  • The authors mention the demonstrated efficacy of HD IL2 in renal cell carcinoma and the FDA approval for renal cell carcinoma and melanoma. However they design their study on breast cancer cells. Either they should mention previous studies on breast cancer or explain why they chose this model.

  • hADSCs is not defined in the introduction (in line 36-37)

  • The authors claim hADSCs are negative for CD11b, CD19, CD34, CD45, and HLA-DR, but they do not show this data in any figure. Provide data.

  • Figure 1c show a WB for IL2. It is not mentioned in the legend how many replicates were done (should be at least 3). The author should mention this and provide a barplot with the quantification of the bands in the 3 replicates

  • Line 417-420. IL2 protein secretion was analyzed in conditioned 417 media (CM) harvested from the native hADSCs, hADSCs-BFP or hADSCs-IL2, and was 418 found to be significantly higher in hADSCs-IL2 (2255.7 ± 738.3 pg/ml, p<0.0001) compared 419 to native hADSCs (2.5 ± 0.2 pg/ml) and hADSCs-BFP (2.0 ± 0.06 pg/ml) at 24 hours (Figure 420 1b).The method to measure secretion of IL-2 protein was not described in the text. How can they confirm whether this is soluble IL-2 or EV-bound IL2?

  • Line 481: "Analysis of the presence of nuclear and mitochondrial components showed the presence of mitochondria stained with MitoTracker Green in 23.78 ± 3.7% of CIMVs isolated from native hADSCs, 21.42 ± 1.56% of CIMVs-BFP and 21.36 ± 2.89% of CIMVs-IL2 (Figure 4c). The nuclear component was stained with the Hoechst 33342 vital fluorescent dye, the fluorescence of which was detected in 7.6 ± 0.9% of CIMVs isolated from native hADSCs, in 5.6 ± 1.5% of CIMVs-BFP, in 6.1 ± 2.9% of CIMVs-IL2 (Figure 4d)." Information on staining method should be indicated in figure legend or method part and not in the main text

  • Figure 4g again as above: It is not mentioned in the legend how many replicates were done (should be at least 3). The author should mention this and provide a barplot with the quantification of the bands in the 3 replicates

  • Figure 4. The size of CIMVs was analysed using flow cytometry.
    Since conventional FACS machines are unable to detect particles below 200 nm, was there any special configuration being applied to detect exosome-like particles (below 100 nm)? Gating strategies including controls, such as PBS and unconditioned media, would give more confidence in separating EVs and other debris or machine noise.

  • Figure 5: the author claim they analyzed the CIMVs with FACS. How did they do that? Common cytometers, including FACSAria III do not have the ability to resolve particles smaller than 500nm. If they claim they can, they should show the facs plots indicating the gating strategy, and use reference-size beads as controls to demonstrate they can resolute the indicated particle size. They might be looking only at a specific subpopulation of the ones indicated in figure 4a (for example >450nm) and this should be clarified to avoid misinterpretation of the data.

  • Figure 5. Immunophenotype of native hADSCs (a), hADSCs-BFP (b) hADSCs-IL2 (c) and CIMVs 518 isolated from each cell line
    What is the negative control used

  • Figure 6: How many events were acquired for each sample (should be indicated in the methods or figure legend)? It is confusing that the author decided to present the data as relative % change between condition. FACS data are usually shown as absolute percentages. Is there a reason for this choice?

  • Line 647 3.6. CIMV-IL2-activated T-cells can kill human triple negative breast cancer cells.
    What is the rationale of choosing human triple negative breast cancer cells? Is this cytotoxic effect from CIMV-IL2-activated T cells only restricted in this cancer type, might be worth to show in other cancer cell lines.

  • Line 714: "In conclusion, the immunomodulating properties of CIMVs-IL2 were not observed in mice." The author show no difference in the number of CD8 and CD4. However, there might be an effect on cytokines production. They should limit their conclusion to the “number” effect and not “immunomodulating” effects. Also, in vitro they saw differences in cd8 killer T cells and they did not look at the overall CD8 population, while now they look at the overall CD8 population. Why? The difference might be only in CD3+ CD8+ CD4- CD38+ HLA-DR+. Also, why didn’t they check other populations like TH17, which showed differences in vitro?

  • "Most commonly, the size of natural EVs isolated from MSCs ranges between 60 nm and 150 nm, corresponding to exosome-like vesicles, with only a small number of EVsranging in size from 200 nm to 400 nm, corresponding to larger EVs [38,39]. The majorityof isolated CIMVs (about 70%) had a size of less than 220 nm, which again corresponds tothe size of the smaller exosome-like vesicles produced endogenously." In this statement the authors contradict themselves: they say endogenous exosomes-like vesicles mostly range between 60 and 150nm. And then they say the CIMVs resemble the endogenous ones since they have 220 nm.

  • The authors should consider comparing their EV related protocols and characterization approach to the Minimal information for studies of extracellular vesicles 2018 (MISEV2018) guidelines for other readers to reference or reproduce the data.

Author Response

Thank you for your comments which have helped us to improve our manuscript. We have corrected your comments point by point within the manuscript accordingly (your comments are in bold text and our responses are in ordinary type):

1) Line 56-57: "The anticancer effect of high dose (HD) IL2 therapy was demonstrated by West et al. where 3 complete and 5 partial responses were reported in patients with metastatic renal cell carcinoma [2]." Out of how many?

Two complete and five partial responses were reported in 23 patients. The appropriate information was added in the manuscript (Page 2, Line 57).

2) The authors mention the demonstrated efficacy of HD IL2 in renal cell carcinoma and the FDA approval for renal cell carcinoma and melanoma. However they design their study on breast cancer cells. Either they should mention previous studies on breast cancer or explain why they chose this model.

Breast cancer is the most commonly diagnosed cancer and the second leading cause of cancer death among women worldwide [72]. One of the challenges in the treatment of breast cancer is tumour heterogeneity, which determines the treatment options [73]. TNBC is the most aggressive and difficult to treat breast cancer subtype, since hormone receptors (HR) are absent and the human epidermal growth factor receptor 2 (HER2) protein is not overexpressed on the surface of TNBC cells. The lack of HR and HER2 expression prevents the use of targeted therapies that are effectively applied to other breast cancer subtypes. Leaving chemotherapy as the only approved option for systemic TNBC treatment. However, the frequency of relapses and metastases of triple-negative breast cancer is very high, and the mean overall survival rate for patients with metastatic TNBC is about 9–12 months, when treated with conventional cytotoxic agents [74]. For TNBC therapy, IL2 was used only in combination with other therapeutic agents, so we decided to analyze its therapeutic potential separately. We have added relevant information in the Discussion to make the choice of MDA-MB-231 cells more reasoned for the reader (Page 26, Line 969-980).

3) hADSCs is not defined in the introduction (in line 36-37)

It was corrected.

4) The authors claim hADSCs are negative for CD11b, CD19, CD34, CD45, and HLA-DR, but they do not show this data in any figure. Provide data.

Information about CD11b, CD19, CD34, CD45, and HLA-DR is presented in Figure 5 as a negative control. This is because in the BD Stemflow™ Human MSC Analysis Kit, a mixture of antibodies against these receptors is presented as a negative cocktail, and staining with these antibodies occurs simultaneously and all the antibodies have the same fluorescent label, so we cannot present information separately for each antigen.

5) Figure 1c show a WB for IL2. It is not mentioned in the legend how many replicates were done (should be at least 3). The author should mention this and provide a barplot with the quantification of the bands in the 3 replicates

Unfortunately, we were not able to use recombinant protein standards during Western blot analysis, so we decided to determine IL2 concentration per µg of total protein in hADSCs-IL2 and CIMVs-IL2 by ELISA in order to support our Western blot results. Methods (Page 8, Lines 314-321) and results (Page 11, Line 475; Page 15, Lines 554-555) were added to the manuscript.

6) Line 417-420. IL2 protein secretion was analyzed in conditioned 417 media (CM) harvested from the native hADSCs, hADSCs-BFP or hADSCs-IL2, and was 418 found to be significantly higher in hADSCs-IL2 (2255.7 ± 738.3 pg/ml, p<0.0001) compared 419 to native hADSCs (2.5 ± 0.2 pg/ml) and hADSCs-BFP (2.0 ± 0.06 pg/ml) at 24 hours (Figure 420 1b).The method to measure secretion of IL-2 protein was not described in the text. How can they confirm whether this is soluble IL-2 or EV-bound IL2?

Secreted protein in CM was measured using cytokine multiplex analysis (described in Materials and Methods section). This method analyzes secreted soluble proteins, but cannot detect proteins inside the vesicles.

7) Line 481: "Analysis of the presence of nuclear and mitochondrial components showed the presence of mitochondria stained with MitoTracker Green in 23.78 ± 3.7% of CIMVs isolated from native hADSCs, 21.42 ± 1.56% of CIMVs-BFP and 21.36 ± 2.89% of CIMVs-IL2 (Figure 4c). The nuclear component was stained with the Hoechst 33342 vital fluorescent dye, the fluorescence of which was detected in 7.6 ± 0.9% of CIMVs isolated from native hADSCs, in 5.6 ± 1.5% of CIMVs-BFP, in 6.1 ± 2.9% of CIMVs-IL2 (Figure 4d)." Information on staining method should be indicated in figure legend or method part and not in the main text

The excessive information was removed.

8) Figure 4g again as above: It is not mentioned in the legend how many replicates were done (should be at least 3). The author should mention this and provide a barplot with the quantification of the bands in the 3 replicates

As mentioned above, we analyzed the concentration of IL2 in CIMVs-IL2 per µg of total protein using ELISA (Page 15, Lines 554-555).

9) Figure 4. The size of CIMVs was analysed using flow cytometry.

Since conventional FACS machines are unable to detect particles below 200 nm, was there any special configuration being applied to detect exosome-like particles (below 100 nm)? Gating strategies including controls, such as PBS and unconditioned media, would give more confidence in separating EVs and other debris or machine noise.

We used a violet laser with a wavelength of 405 nm as FSC instead of 488 nm laser since shorter wavelength is better suited to identify particles with smaller diameter. The use of violet laser (Ex=405, Em=450) allows detecting particles above 200 nm in size. We added appropriate information into the text (Page 7, Lines 275-276).

10) Figure 5: the author claim they analyzed the CIMVs with FACS. How did they do that? Common cytometers, including FACSAria III do not have the ability to resolve particles smaller than 500nm. If they claim they can, they should show the facs plots indicating the gating strategy, and use reference-size beads as controls to demonstrate they can resolute the indicated particle size. They might be looking only at a specific subpopulation of the ones indicated in figure 4a (for example >450nm) and this should be clarified to avoid misinterpretation of the data.

As described above we used violet laser (Ex=405, Em=450) to detected the particles which are more than 200 nm in size.

11) Figure 5. Immunophenotype of native hADSCs (a), hADSCs-BFP (b) hADSCs-IL2 (c) and CIMVs 518 isolated from each cell line

What is the negative control used

Negative control is negative cocktail from BD Stemflow™ Human MSC Analysis Kit which contains PE-conjugated anti-CD11b, anti-CD19, anti-CD34, anti-CD45, and anti-HLA-DR antibodies.

12) Figure 6: How many events were acquired for each sample (should be indicated in the methods or figure legend)? It is confusing that the author decided to present the data as relative % change between condition. FACS data are usually shown as absolute percentages. Is there a reason for this choice?

A minimum of 20,000 events were acquired for each PBMC sample, we have included this information in each Materials and Methods subsection where it is required. Data were presented as relative to control since significant range of the number of cells in populations was observed in different donors. Therefore, we decided to calculate the data as a relative to the control.

13) Line 647 3.6. CIMV-IL2-activated T-cells can kill human triple negative breast cancer cells.

What is the rationale of choosing human triple negative breast cancer cells? Is this cytotoxic effect from CIMV-IL2-activated T cells only restricted in this cancer type, might be worth to show in other cancer cell lines.

The reason for choosing human triple negative breast cancer cells is described above in answer to the question 2. The therapeutic effect of CIMVs-IL2 on other tumor models is going to be investigated in the further studies.

14) Line 714: “In conclusion, the immunomodulating properties of CIMVs-IL2 were not observed in mice.” The author show no difference in the number of CD8 and CD4. However, there might be an effect on cytokines production. They should limit their conclusion to the “number” effect and not “immunomodulating” effects. Also, in vitro they saw differences in cd8 killer T cells and they did not look at the overall CD8 population, while now they look at the overall CD8 population. Why? The difference might be only in CD3+ CD8+ CD4- CD38+ HLA-DR+. Also, why didn’t they check other populations like TH17, which showed differences in vitro?

The reason for the restricted analysis of mouse cell populations is that we were limited in our choice of anti-mouse antibodies, so we analyzed only CD8 and CD4 populations. However, in order to get a more detailed view and draw conclusions regarding immunomodulation, we followed your advice and analyzed the cytokine profile of mice after the administration of CIMVs. We found an increase in the secretion of pro-inflammatory cytokines in the serum of mice treated with native CIMVs and CIMVs-BFP, but not CIMVs-IL2. The results are described and discussed in Page 21, Line 774 - Page 22, Line 811 and Page 26, Lines 1012-1039, respectively.

15) “Most commonly, the size of natural EVs isolated from MSCs ranges between 60 nm and 150 nm, corresponding to exosome-like vesicles, with only a small number of Evsranging in size from 200 nm to 400 nm, corresponding to larger EVs [38,39]. The majorityof isolated CIMVs (about 70%) had a size of less than 220 nm, which again corresponds to the size of the smaller exosome-like vesicles produced endogenously.” In this statement the authors contradict themselves: they say endogenous exosomes-like vesicles mostly range between 60 and 150nm. And then they say the CIMVs resemble the endogenous ones since they have 220 nm.

We were able to detect particles above 200 nm using violet laser. Based on the size of the calibration particles, it was shown that most of the CIMVs were smaller than 220 nm in size.

16) The authors should consider comparing their EV related protocols and characterization approach to the Minimal information for studies of extracellular vesicles 2018 (MISEV2018) guidelines for other readers to reference or reproduce the data.

Since our vesicles are not endogenous, but were isolated mechanically using Cytochalasin B, we tried to analyze if they retain the characteristics of parental stem cells. At the same time, we tried to rely on the MISEV2018 guide and showed the presence of transmembrane proteins (MSC CD markers) on the surface of the vesicles and cytosolic proteins (IL2). In addition, we carried out single vesicle analysis (scanning electron microscopy, flow cytometry), and produced information on the content of mitochondrial and nuclear components in CIMVs. Since our CIMVs appear to differ from endogenous EVs, we are planning to conduct more detailed comparison study in the future.

Round 2

Reviewer 1 Report

I am happy to see the author’s response to my comments. I really appreciate their time and work but still, we may not able to conclude this work with only one TNBC cells (MDA-MB-231 cells). So, if possible, including Annexin V Apoptosis assay for another TNBC cell line (such as MDA-MB-436, 468) may strengthen the value of this manuscript.   

Author Response

Thank you for your comments which have helped us to improve our manuscript. We analyzed cytotoxic effect of PBMCs on MDA-MB-436 cell line. Methods are described in Page 9, Lines 395-403. Results are described and discussed in Page 21, Lines 752-768 and Page 27, Lines 1013-1017, respectively.

Reviewer 2 Report

We believe the authors have satisfactorily addressed most of our comments and we are happy to endorse this manuscript now.

Author Response

Thank you for your comments. We additionally analyzed cytotoxic effect of PBMCs on MDA-MB-436 cell line. Methods are described in Page 9, Lines 395-403. Results are described and discussed in Page 21, Lines 752-768 and Page 27, Lines 1013-1017, respectively.